# Inherently Interpretable Time Series Classification via Multiple Instance Learning

**Joseph Early,*  Gavin KC Cheung,† Kurt Cutajar,† Hanting Xie,† Jas Kandola,† & Niall Twomey†**
Corresponding authors: `J.A.Early@soton.ac.uk`; `njtwomey@amazon.co.uk`

## Abstract

Conventional Time Series Classification (TSC) methods are often *black boxes* that obscure inherent interpretation of their decision-making processes. In this work, we leverage Multiple Instance Learning (MIL) to overcome this issue, and propose a new framework called **MILLET: M**ultiple **I**nstance **L**earning for **L**ocally **E**xplainable **T**ime series classification. We apply `MILLET` to existing deep learning TSC models and show how they become inherently interpretable without compromising (and in some cases, even improving) predictive performance. We evaluate `MILLET` on 85 UCR TSC datasets and also present a novel synthetic dataset that is specially designed to facilitate interpretability evaluation. On these datasets, we show `MILLET` produces sparse explanations quickly that are of higher quality than other well-known interpretability methods. To the best of our knowledge, our work with `MILLET`, which is available on GitHub[1], is the first to develop general MIL methods for TSC and apply them to an extensive variety of domains.

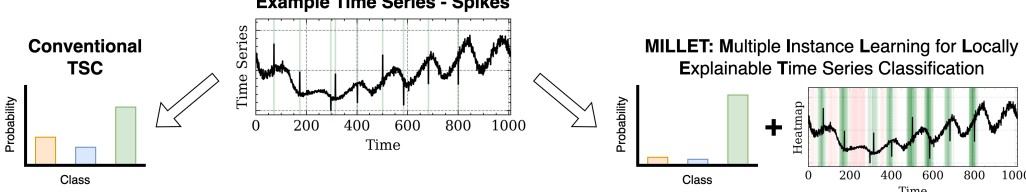

Figure 1: Conventional TSC techniques (left) usually only provide class-level predictive probabilities. In addition, our proposed method (`MILLET`, right) also highlights class-conditional discriminatory motifs that influence the predicted class. In the heatmap, green regions indicate support for the predicted class, red regions refute the predicted class, and darker regions are more influential.

## 1 Introduction

Time Series Classification (TSC) is the process of assigning labels to sequences of data, and occurs in a wide range of settings – examples from the popular UCR collection of datasets include predicting heart failure from electrocardiogram data, and identifying household electric appliance usage from electricity data (Dau et al., 2019). Each of these domains have their own set of class-conditional discriminatory motifs (the signatures that determine the class of a time series). Deep Learning (DL) methods have emerged as a popular family of approaches for solving TSC problems. However, we identify two drawbacks with these conventional supervised learning approaches: 1) representations are learnt for each time point in a time series, but these representations are then lost through an aggregation process that weights all time points equally, and 2) these methods are *black boxes* that provide no inherent explanations for their decision making, i.e. they cannot localise the class-conditional discriminatory motifs. These drawbacks not only limit predictive performance, but also introduce barriers to their adoption in practice as the models are not transparent.

To mitigate these shortcomings, we take an alternative view of DL for TSC, approaching it as a Multiple Instance Learning (MIL) problem. MIL is a weakly supervised learning paradigm in which a collection (*bag*) of elements (*MIL instances*) all share the same label. In the context of TSC, a bag is a time series of data over a contiguous interval. In the MIL setting, the learning objective is to assign class labels to unlabelled bags of time series data whilst also discovering the salient motifs

---

*University of Southampton, UK – work completed during an internship at Amazon Prime Video, UK.
†Amazon Prime Video, UK

[1]`https://github.com/JAEarly/MILTimeSeriesClassification`

within the time series that explain the reasons for the predicted class. As we explore in this work, MIL is well-suited to overcome the drawbacks identified above, leading to inherent interpretability without compromising predictive performance (even improving it in some cases). We propose a new general framework applying MIL to TSC called **MILLET: M**ultiple **I**nstance **L**earning for **L**ocally **E**xplainable **T**ime series classification. Demonstrative `MILLET` model outputs are depicted in Fig. 1.

MIL is well-suited to this problem setting since it was developed for weakly supervised contexts, can be learnt in an end-to-end framework, and boasts many successes across several domains. Furthermore, MIL has the same label specificity as TSC: labels are given at the bag level, but not at the MIL instance level. To explore the intersection of these two areas, we propose *plug-and-play* concepts that are adapted from MIL and applied to existing TSC approaches (in this work, DL models[2]). Furthermore, to aid in our evaluation of the interpretability of these new methods, we introduce a new synthetic TSC dataset, *WebTraffic*, where the location of the class-conditional discriminatory motifs within time series are known. The time series shown in Fig. 1 is sampled from this dataset.

Our key contributions are as follows:

1. We propose `MILLET`, a TSC framework that utilises MIL to provide inherent interpretability without compromising predictive performance (even improving it in some cases).
2. We design *plug-and-play* MIL methods for TSC within `MILLET`.
3. We propose a new method of MIL aggregation, `Conjunctive pooling`, that outperforms existing pooling methods in our TSC experiments.
4. We propose and evaluate 12 novel `MILLET` models on 85 univariate datasets from the UCR TSC Archive (Dau et al., 2019), as well as a novel synthetic dataset that facilitates better evaluation of TSC interpretability.

## 2 BACKGROUND AND RELATED WORK

**Time Series Classification**   While a range of TSC methods exist, in this work we apply MIL to DL TSC approaches. Methods in this family are effective and widely used (Ismail Fawaz et al., 2019; Foumani et al., 2023); popular methods include Fully Convolutional Networks (`FCN`), Residual Networks (`ResNet`), and `InceptionTime` (Wang et al., 2017; Ismail Fawaz et al., 2020). Indeed, a recent TSC survey, *Bake Off Redux* (Middlehurst et al., 2023), found `InceptionTime` to be competitive with SOTA approaches such as the ensemble method HIVE-COTE 2 (`HC2`; Middlehurst et al., 2021) and the hybrid dictionary-convolutional method Hydra-MultiRocket (`Hydra-MR`; Dempster et al., 2023). Although the application of Matrix Profile for TSC also yields inherent interpretability (Yeh et al., 2017; Guidotti & D'Onofrio, 2021), we choose to focus on DL approaches due to their popularity, strong performance, and scope for improvement (Middlehurst et al., 2023).

**Multiple Instance Learning**   In its standard assumption, MIL is a binary classification problem: a bag is positive if and only if at least one of its instances is positive (Dietterich et al., 1997). As we are designing `MILLET` to be a general and widely applicable TSC approach, we do not constrain it to any specific MIL assumption except that there are temporal relationships, i.e. the order of instances within bags matters (Early et al., 2022; Wang et al., 2020). As we explore in Sec. 3.4, this allows us to use positional encodings in our `MILLET` methods. Although the application of MIL to TSC has been explored prior to this study, earlier work focused on domain-specific problems such as intensive care in medicine and human activity recognition (Dennis et al., 2018; Janakiraman, 2018; Poyiadzi et al., 2018; Poyiadzis et al., 2019; Shanmugam et al., 2019). Furthermore, existing work considers MIL as its own unique approach separate from existing TSC methods. The work most closely related to ours is Zhu et al. (2021), which proposes an uncertainty-aware MIL TSC framework specifically designed for long time series (marine vessel tracking), but without the generality and *plug-and-play* nature of `MILLET`. Therefore, to the best of our knowledge, our work with `MILLET` is the first to apply MIL to TSC in a more general sense and to do so across an extensive variety of domains.

**Interpretability**   TSC interpretability methods can be grouped into several categories (Theissler et al., 2022) – in this work we focus on class-wise time point attribution (saliency maps), i.e. identifying the discriminatory time points in a time series that support and refute different classes. This is a form of local interpretation, where model decision-making is explained for individual time series

---

[2]While we focus on DL TSC in this work, we envision that our `MILLET` framework can be applied to other TSC approaches in the future, such as the `ROCKET` family of methods (Dempster et al., 2020; 2023).

(Molnar, 2022). It also aligns with MIL interpretability as proposed by Early et al. (2021): *which* are the key MIL instances in a bag, and *what* outcomes do they support/refute? MILLET facilitates interpretability by inherently enhancing existing TSC approaches such that they provide interpretations alongside their predictions with a single forward pass of the model. This is in contrast to perturbation methods such as LIME (Ribeiro et al., 2016), SHAP (Lundberg & Lee, 2017), Occlusion Sensitivity (Zeiler & Fergus, 2014), and MILLI (Early et al., 2021), which are much more expensive to run (often requiring 100+ forward passes per interpretation). An interpretability approach that can be run with a single forward pass is Class Activation Mapping (CAM) (Zhou et al., 2016; Wang et al., 2017). It uses the model's weights to identify discriminatory time points, and serves as a benchmark in this work. For more details on existing TSC interpretability methods and their evaluation metrics, see App. A.1, Theissler et al. (2022), and Šimić et al. (2021).

## 3 METHODOLOGY

To apply MIL to TSC, we propose the broad framework **MILLET: M**ultiple **I**nstance **L**earning for **L**ocally **E**xplainable **T**ime series classification. We advocate for the use of MIL in TSC as it is a natural fit that provides inherent interpretability (explanations for free) without requiring any additional labelling beyond that already provided by existing TSC datasets (we further discuss our motivation for using MIL in App. A).

### 3.1 THE MILLET FRAMEWORK

A TSC model within our MILLET framework has to satisfy three requirements:

**Requirement 1: Time Series as MIL Bags** Input data consists of time series, $\mathbf{X_i}$, where a time series is formed of $t > 1$ time points: $\mathbf{X_i} = \{\mathbf{x_i^1}, \mathbf{x_i^2}, \ldots, \mathbf{x_i^t}\}$ and $i$ is the sample index.[3] Each time step is a $c$-dimensional vector, where $c$ is the number of channels in the time series – in this work we focus on univariate time series ($c = 1$) and assume all time series in a dataset have the same length. We consider each time series as a MIL bag, meaning each time point is a MIL instance.[4] A time series bag can be denoted as $\mathbf{X_i} \in \mathbb{R}^{t \times c}$, and each bag has an associated bag-level label $Y_i$ which is the original time series label. There is also the concept of MIL instance labels $\{y_i^1, y_i^2, \ldots, y_i^t\}$, but these are not provided for most MIL datasets (like the absence of time point labels in TSC). Framing TSC as a MIL problem allows us to obtain interpretability by imposing the next requirement.

**Requirement 2: Time Point Predictions** To facilitate interpretability in our framework, we specify that models must provide time point predictions along with their time series predictions. Furthermore, the time point predictions should be inherent to the model – this makes it possible to identify which time points support and refute different classes without having to use post-hoc methods.

**Requirement 3: Temporal Ordering** TSC is a sequential learning problem so we impose a further requirement that the framework must respect the ordering of time points. This is in contrast to classical MIL methods that assume MIL instances are iid.

### 3.2 MILLET FOR DL TSC: RETHINKING POOLING

To demonstrate the use of MILLET, we apply it to DL TSC methods. Existing DL TSC architectures (e.g. FCN, ResNet, and InceptionTime) mainly consist of two modules: a feature extractor $\psi_{FE}$ (we refer to these as backbones) and a classifier $\psi_{CLF}$. For an input univariate time series $\mathbf{X_i}$, $\psi_{FE}$ produces a set of $d$-dimensional feature embeddings $\mathbf{Z_i} \in \mathbb{R}^{t \times d} = [\mathbf{z_i^1}, \mathbf{z_i^2}, \ldots, \mathbf{z_i^t}]$. These embeddings are consolidated via aggregation with Global Average Pooling (GAP) to give a single feature vector of length $d$. This is then passed to $\psi_{CLF}$ to produce predictions for the time series:

$$\text{Feature Extraction: } \mathbf{Z_i} = \psi_{FE}(\mathbf{X_i}); \quad \text{GAP + Classification: } \hat{\mathbf{Y}}_\mathbf{i} = \psi_{CLF}\left(\frac{1}{t}\sum_{j=1}^{t} \mathbf{z_i^j}\right). \quad (1)$$

---

[3]Following convention from MIL, we use uppercase variables to denote MIL bag / time series data and lowercase variables to denote MIL instance / time point data.

[4]There is an overlap in TSC and MIL terminology: both use the term 'instance' but in different ways. In MIL it denotes an element in a bag, and in TSC it refers to an entire time series (e.g. "instance-based explanations" from Theissler et al., 2022). To avoid confusion, we use 'time series' to refer to entire time series (a TSC instance) and 'time point' to refer to a value for a particular step in a time series (a MIL instance).

The specification of $\psi_{FE}$ naturally satisfies Req. 1 from our `MILLET` framework as discriminatory information is extracted on a time point level. Req. 3 is satisfied as long as the DL architecture makes use of layers that respect the ordering of the time series such as convolutional or recurrent layers. In the MIL domain, the GAP + Classification process (Eqn. 1) is known as mean `Embedding` pooling, as used in methods such as MI-Net (Wang et al., 2018). However, this aggregation step does not inherently produce time point class predictions, and consequently does not fit Req. 2.

To upgrade existing DL TSC methods into the `MILLET` framework and satisfy Req. 2, we explore four MIL pooling methods for replacing GAP. `Attention`, `Instance`, and `Additive` are inspired by existing MIL approaches, while `Conjunctive` is proposed in this work. Replacing GAP in this way is *plug-and-play*, i.e. any TSC method using GAP or similar pooling can easily be upgraded to one of these methods and meet the requirements for `MILLET`.

**Attention pooling** (Ilse et al., 2018) does weighted averaging via an attention head $\psi_{ATTN}$:

$$a_i^j \in [0,1] = \psi_{ATTN}\left(\mathbf{z_i^j}\right); \qquad \hat{\mathbf{Y}}_\mathbf{i} = \psi_{CLF}\left(\frac{1}{t}\sum_{j=1}^{t} a_i^j \mathbf{z_i^j}\right). \tag{2}$$

**Instance pooling** (Wang et al., 2018) makes a prediction for each time point:

$$\hat{\mathbf{y}}_\mathbf{i}^\mathbf{j} \in \mathbb{R}^c = \psi_{CLF}\left(\mathbf{z_i^j}\right); \qquad \hat{\mathbf{Y}}_\mathbf{i} = \frac{1}{t}\sum_{j=1}^{t}\left(\hat{\mathbf{y}}_\mathbf{i}^\mathbf{j}\right). \tag{3}$$

**Additive pooling** (Javed et al., 2022) is a combination of `Attention` and `Instance`:

$$a_i^j \in [0,1] = \psi_{ATTN}\left(\mathbf{z_i^j}\right); \qquad \hat{\mathbf{y}}_\mathbf{i}^\mathbf{j} = \psi_{CLF}\left(a_i^j \mathbf{z_i^j}\right); \qquad \hat{\mathbf{Y}}_\mathbf{i} = \frac{1}{t}\sum_{j=1}^{t}\left(\hat{\mathbf{y}}_\mathbf{i}^\mathbf{j}\right). \tag{4}$$

**Conjunctive pooling** is our proposed novel pooling approach, where attention and classification are independently applied to the time point embeddings, after which the attention values are used to scale the time point predictions. This is expected to benefit performance as the attention and classifier heads are trained in parallel rather than sequentially, i.e. the classifier cannot rely on the attention head to alter the time point embeddings prior to classification, making it more robust. We use the term `Conjunctive` to emphasise that, from an interpretability perspective, a discriminatory time point must be considered important by both the attention head *and* the classification head. Formally, `Conjunctive` is described as:

$$a_i^j \in [0,1] = \psi_{ATTN}\left(\mathbf{z_i^j}\right); \qquad \hat{\mathbf{y}}_\mathbf{i}^\mathbf{j} = \psi_{CLF}\left(\mathbf{z_i^j}\right); \qquad \hat{\mathbf{Y}} = \frac{1}{t}\sum_{j=1}^{t}\left(a_i^j \hat{\mathbf{y}}_\mathbf{i}^\mathbf{j}\right). \tag{5}$$

Fig. 2 shows a schematic representation comparing these pooling approaches with `Embedding` (GAP). Note there are other variations of these MIL pooling methods, such as replacing mean with max in `Embedding` and `Instance`, but these alternative approaches are not explored in this work.

### 3.3 MILLET DL INTERPRETABILITY

As a result of Requirement 2 in Sec. 3.1, we expect the models to be inherently interpretable. For DL methods, this is achieved through the MIL pooling methods given in Sec. 3.2 – different MIL pooling approaches provide alternative forms of inherent interpretability. `Instance` performs classification before pooling (see Eqn. 3), so it produces a set of time point predictions $\hat{\mathbf{y}}_\mathbf{i} \in \mathbb{R}^{t \times c} = [\hat{\mathbf{y}}_\mathbf{i}^\mathbf{1}, \hat{\mathbf{y}}_\mathbf{i}^\mathbf{2}, \ldots, \hat{\mathbf{y}}_\mathbf{i}^\mathbf{t}]$. `Additive` and `Conjunctive` also make time point predictions, but include attention. To combine these two outputs, we weight the time point predictions by the attention scores: $\hat{\mathbf{y}}_\mathbf{i}^* \in \mathbb{R}^{t \times c} = [a_i^1 \hat{\mathbf{y}}_\mathbf{i}^\mathbf{1}, a_i^2 \hat{\mathbf{y}}_\mathbf{i}^\mathbf{2}, \ldots, a_i^t \hat{\mathbf{y}}_\mathbf{i}^\mathbf{t}]$. Note that $\hat{\mathbf{y}}_\mathbf{i}^*$ is used to signify the attention weighting of the original time point predictions $\hat{\mathbf{y}}_\mathbf{i}$.

On the other hand, `Attention` is inherently interpretable through its attention weights $\mathbf{a_i} \in [0,1]^t = [a_i^1, a_i^2, \ldots, a_i^t]$, which can be interpreted as a measure of importance for each time point. Note, unlike `Instance`, `Additive`, and `Conjunctive`, the interpretability output for `Attention` is not class specific (but only a general measure of importance across all classes).

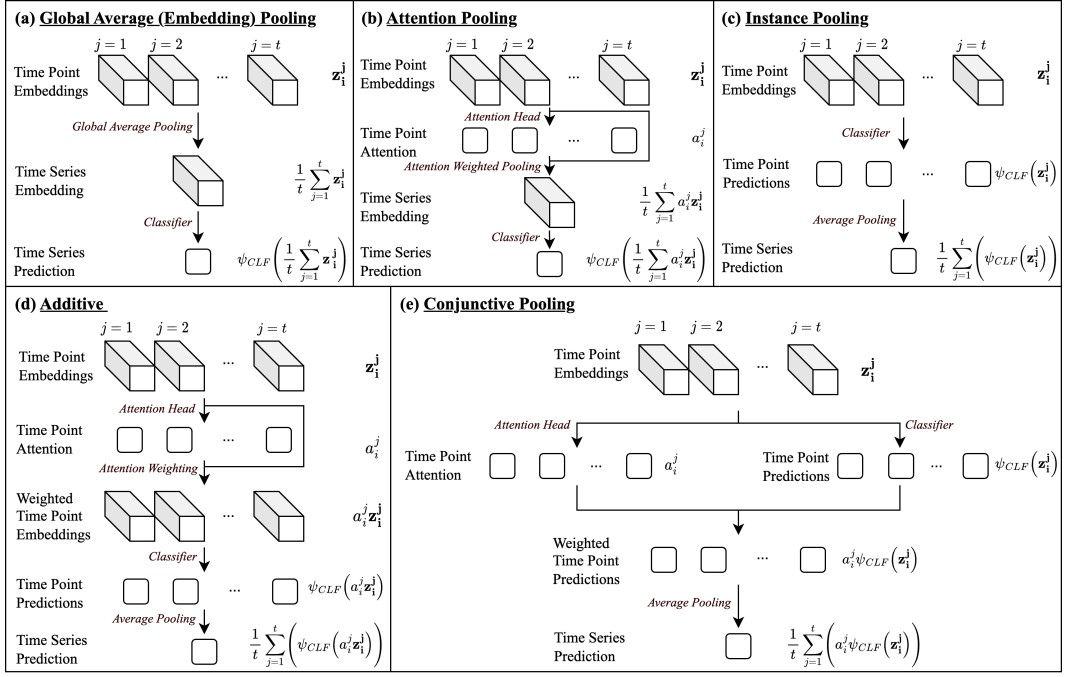

Figure 2: The five different MIL pooling methods used in this work. Each takes the same input: a bag of time point embeddings $\mathbf{Z_i} \in \mathbb{R}^{t \times d} = [\mathbf{z_i^1}, \mathbf{z_i^2}, \dots, \mathbf{z_i^t}]$. While they all produce the same overall output (a time series prediction), they produce different interpretability outputs.

## 3.4  MILLET DL MODEL DESIGN

We design three MILLET DL models by adapting existing backbone models that use GAP: FCN, ResNet, and InceptionTime. While extensions of these methods and other DL approaches exist (see Foumani et al., 2023), we do not explore these as none have been shown to outperform InceptionTime (Middlehurst et al., 2023). Nevertheless, the MILLET framework can be applied to any generic DL TSC approach that uses GAP or follows the high-level structure in Eqn. 1.

Replacing GAP with one of the four pooling methods in Sec. 3.2 yields a total of 12 new models. In each case, the backbone models produce feature embeddings of length $d = 128$ $\left(\mathbf{Z_i} \in \mathbb{R}^{t \times 128}\right)$. The models are trained end-to-end in the same manner as the original backbone methods – we discuss additional options for training in Sec. 6. We introduce three further enhancements:

1. **Positional Encoding:** As time point classification and attention are applied to each time point independently, the position of a time point within the times series can be utilised (with GAP, positional encoding would be lost through averaging) – this allows for further expressivity of the ordering of time points and enforces Req. 3 of our MILLET framework. We inject fixed positional encodings (Vaswani et al., 2017) after feature extraction.

2. **Replicate padding:** Zero padding is used in the convolutional layers of the backbone architectures. However, in our interpretability experiments, we found this biased the models towards the start and end of the time series – padding with zeros was creating a false signal in the time series. As such, we replaced zero padding with replicate padding (padding with the boundary value) which alleviated the start/end bias. However, we note that particular problems may benefit from other padding strategies.

3. **Dropout:** To mitigate overfitting in the new pooling methods, we apply dropout after injecting the positional encodings ($p = 0.1$). No dropout was used in the original backbones.

The original InceptionTime approach is an ensemble of five identical network architectures trained from different initialisations, where the overall output is the mean output of the five networks. To facilitate a fair comparison, we use the same approach for FCN, ResNet, and MILLET. See Sec. 6 for implementation details, and App. B for model, training, and hyperparameter details.

## 4 INITIAL CASE STUDY

### 4.1 SYNTHETIC DATASET

In order to explore our `MILLET` concept, and to evaluate the inherent interpretability of our models, we propose a new synthetic dataset called *WebTraffic*. By demonstrating daily and weekly seasonality, it is designed to mimic trends observed in streaming and e-commerce platforms. We inject different signatures into a collection of synthetic time series to create ten different classes: a zeroth normal class and nine signature classes. The signatures are partially inspired by the synthetic anomaly types proposed in Goswami et al. (2023). The discriminatory time points are known as we are able to control the location of the injected signatures. Therefore, we are able to evaluate if models can identify both the signature and the location of the discriminatory time points – both can be achieved inherently by our `MILLET` models. Each time series is of length $t = 1008$, and the training and test set contain 500 time series – see Fig. 3 for examples and App. C.1 for more details.

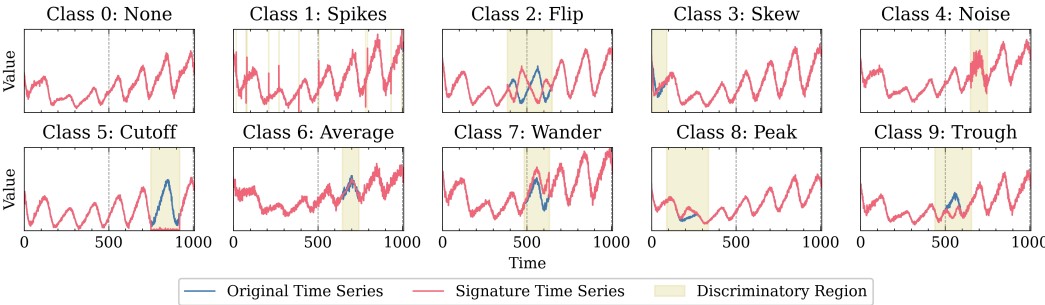

Figure 3: An example for each class of our *WebTraffic* dataset. Signatures are injected in a single random window, with the exception of Class 1 (Spikes), which uses random individual time points.

### 4.2 *WebTraffic* RESULTS

We compare the four proposed MIL pooling approaches for `MILLET` with GAP on our *WebTraffic* dataset. Each pooling method is applied to the `FCN`, `ResNet`, and `InceptionTime` backbones. We conduct five training repeats of each model, starting from different network initialisations, and then ensemble them to a single model. We find that `MILLET` improves interpretability without being detrimental to predictive performance. In actuality, `MILLET` improves accuracy averaged across all backbones from 0.850 to 0.874, with a maximum accuracy of 0.940 for `Conjunctive InceptionTime`.[5] In Fig. 4 we give example interpretations for this best-performing model. The model is able to identify the correct discriminatory regions for the different classes, but focuses on certain parts of the different signatures. For example, for the Spikes class, the model identifies regions surrounding the individual spike time points, and for the Cutoff class, the model mainly identifies the start and end of the discriminatory region. This demonstrates how our interpretability outputs are not only able to convey where the discriminatory regions are located, but also provide insight into the model's decision-making process, providing transparency.

To quantitatively evaluate interpretability on our *WebTraffic* dataset, we use the same process as Early et al. (2021). This approach uses ranking metrics, i.e. looking at the predicted importance order rather than actual interpretation values. The two metrics used are Area Over The Perturbation Curve to Random (AOPCR) and Normalised Discounted Cumulative Gain at $n$ (NDCG@n). The former is used to evaluate without time point labels, and the latter is used to evaluate with time point labels.[6] In this evaluation, we compare to baselines CAM (applied to the original GAP models) and SHAP (applied to all models, see App. B.3). CAM is a lightweight post-hoc interpretability method (but not intrinsically part of the model output unlike our `MILLET` interpretations). SHAP is much more expensive to run than CAM or `MILLET` as it has to make repeated forward passes of the model. In this case, we use SHAP with 500 samples, meaning it is 500 times more expensive than `MILLET`. In actuality, we find `MILLET` is over 800 times faster than SHAP (see App. E.4).

---

[5]For complete results, see App. D.2.

[6]For more details on both metrics, see App. D.1. Note that NDCG@n can be used for this dataset as we know the locations of discriminatory time points (where the signatures were injected). However, for the UCR datasets used in Sec. 5, the discriminatory time points are unknown, therefore only AOPCR can be used.

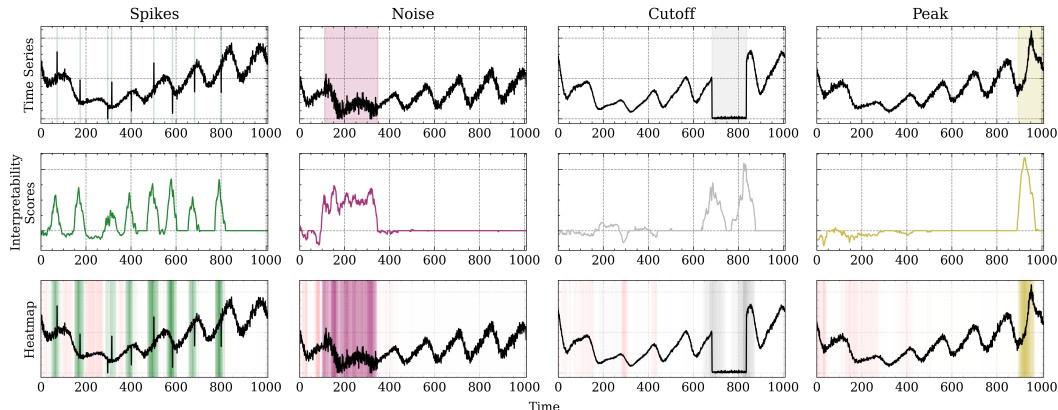

Figure 4: Interpretations for `Conj. InceptionTime` on our *WebTraffic* dataset. **Top:** Time series with the known discriminatory time points highlighted. **Middle:** Interpretability scores for each time point with respect to the target class. **Bottom:** Interpretability scores heatmap as in Fig. 1.

As shown in Table 1, `MILLET` provides better interpretability performance than CAM or SHAP. SHAP performs particularly poorly, especially considering it is so much more expensive to run. Due to the exponential number of possible coalitions, SHAP struggles with the large number of time points. In some cases, it even has a negative AOPCR score, meaning its explanations are worse than random. For each backbone, `MILLET` has the best AOPCR and NDCG@n performance. The exception to this is NDCG@n for `InceptionTime`, where CAM is better (despite `MILLET` having a better AOPCR score). This is likely due to the sparsity of `MILLET` explanations – as shown for the Cutoff and Peak examples in Fig. 4, `MILLET` produces explanations that may not achieve full coverage of the discriminatory regions. While sparsity is beneficial for AOPCR (fewer time points need to be removed to decay the prediction), it can reduce NDCG@n as some discriminatory time points may not be identified (for example those in the middle of the Cutoff region).

Table 1: Interpretability performance (AOPCR / NDCG@n) on *WebTraffic*. For SHAP and `MILLET`, results are given for the best performing pooling method. For complete results, see App. D.2.

|  | FCN | ResNet | InceptionTime | Mean |
|---|---|---|---|---|
| CAM | 12.780 / 0.532 | 20.995 / 0.582 | 12.470 / **0.707** | 15.415 / 0.607 |
| SHAP | 1.977 / 0.283 | -0.035 / 0.257 | -4.020 / 0.259 | -0.692 / 0.266 |
| MILLET | **14.522** / **0.540** | **24.880** / **0.591** | **13.192** / 0.704 | **17.531** / **0.612** |

## 5 UCR RESULTS

We evaluate `MILLET` on the UCR TSC Archive (Dau et al., 2019) – widely acknowledged as a definitive TSC benchmark spanning diverse domains across 85 univariate datasets (see App. C.2). Below are results for predictive performance, followed by results for interpretability.

### 5.1 PREDICTIVE PERFORMANCE

For the three backbones, we compare the performance of the four MIL pooling methods with that of GAP. This is used to evaluate the change in predictive performance when using `MILLET` in a wide variety of different domains, and determine which of the pooling methods proposed in Sec. 3.2 is best. Averaged across all backbones, we find `Conjunctive` gives the best performance, with an accuracy improvement from $0.841 \pm 0.009$ to $0.846 \pm 0.009$ when compared to GAP. `Conjunctive InceptionTime` has the highest average accuracy of $0.856 \pm 0.015$ (see App. D.3 for all results). Given this result, we then compare the performance of the three `MILLET` `Conjunctive` approaches with current SOTA methods, which is intended to provide better context for the performance of our newly proposed models. We select the top performing method from seven families of TSC approaches as outlined by Middlehurst et al. (2023).

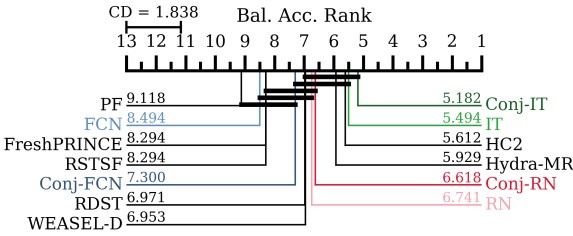

Figure 5: Critical difference diagram comparing Conjunctive MILLET methods with SOTA.

As HC2 is a very computationally expensive meta-ensemble of multiple different classifiers, we do not consider it to be an equal comparison to our methods. In Fig. 5 we give a critical difference (CD) diagram (Demšar, 2006) for balanced accuracy. We find that 1) using Conjunctive improves performance in all cases, and 2) Conjunctive InceptionTime is comparable to (even slightly better than) the SOTA of HC2 and Hydra-MR. We provide further results in Table 2. While Conjunctive InceptionTime is the best approach on balanced accuracy (outperforming the HC2 and Hydra-MR SOTA methods), it is not quite as strong on the other metrics. However, it remains competitive, and for each backbone using MILLET improves performance across all metrics.

Table 2: Performance on 85 UCR datasets in the form: mean / rank / number of wins. HC2 is included for reference but not compared to as it is a meta-ensemble of several other TSC approaches.

| Method | Accuracy $^\uparrow$ | Bal. Accuracy $^\uparrow$ | AUROC $^\uparrow$ | NLL $^\downarrow$ |
|---|---|---|---|---|
| Hydra-MR | **0.857** / **5.306** / **19** | 0.831 / 5.929 / 17 | 0.875 / 10.682 / 7 | **0.953** / 6.965 / **9** |
| FCN | 0.828 / 9.088 / 7 | 0.804 / 8.494 / 8 | 0.929 / 7.653 / 13 | 1.038 / 7.176 / 5 |
| Conj. FCN | 0.838 / 7.700 / 7 | 0.814 / 7.300 / 7 | 0.934 / 6.288 / 20 | 0.973 / **6.247** / **9** |
| ResNet | 0.843 / 7.282 / 10 | 0.819 / 6.741 / 8 | 0.937 / 6.035 / 18 | 1.091 / 7.788 / 0 |
| Conj. ResNet | 0.845 / 7.200 / 13 | 0.822 / 6.618 / 14 | **0.939** / 5.512 / 16 | 1.035 / 7.447 / 2 |
| ITime | 0.853 / 6.112 / **19** | 0.832 / 5.494 / 19 | **0.939** / 5.453 / **26** | 1.078 / 7.118 / **9** |
| Conj. ITime | 0.856 / 5.606 / **19** | **0.834** / **5.182** / **23** | **0.939** / **5.276** / **26** | 1.085 / 7.341 / **9** |
| *HC2* | *0.860 / 4.953 / 21* | *0.830 / 5.612 / 17* | *0.950 / 3.441 / 43* | *0.607 / 4.941 / 14* |

## 5.2 INTERPRETABILITY PERFORMANCE

In order to understand the interpretability of our MILLET methods on a wide variety of datasets, we evaluate their interpretability on the same set of 85 UCR datasets used in Sec. 5.1.

As the UCR datasets do not have time point labels, we can only evaluate model interpretability using AOPCR. Averaged across all backbones, we find that MILLET has a best AOPCR of 6.00, compared to 5.71 achieved by GAP. Of the individual pooling methods within MILLET, we find that Conjunctive has the best interpretability performance. Attention performs poorly – this is expected as it does not create class-specific interpretations, but only general measures of importance; also identified in general MIL interpretability by Early et al. (2021). In Fig. 6 we observe a trade-off between interpretability and prediction, something we believe is insightful for model selection in practice. As backbone complexity increases, predictive performance increases while interpretability decreases.[7] The Pareto front shows MILLET dominates GAP for FCN and InceptionTime, but not ResNet. MILLET gives better interpretability than GAP for *individual* ResNet models, but struggles with the ensemble ResNet models. For complete results and a further discussion, see App. D.3.

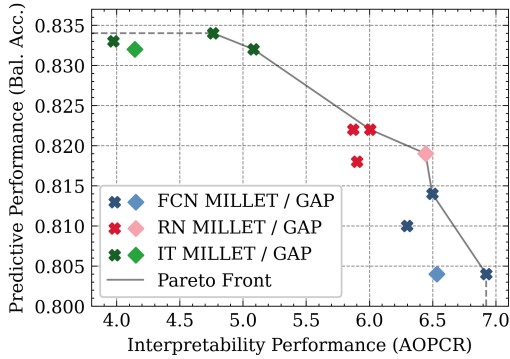

Figure 6: The interpretability-predictive performance trade-off. Each cross indicates a different pooling method (Attention omitted).

---

[7] InceptionTime is the most complex backbone, followed by ResNet, and then FCN is the simplest.

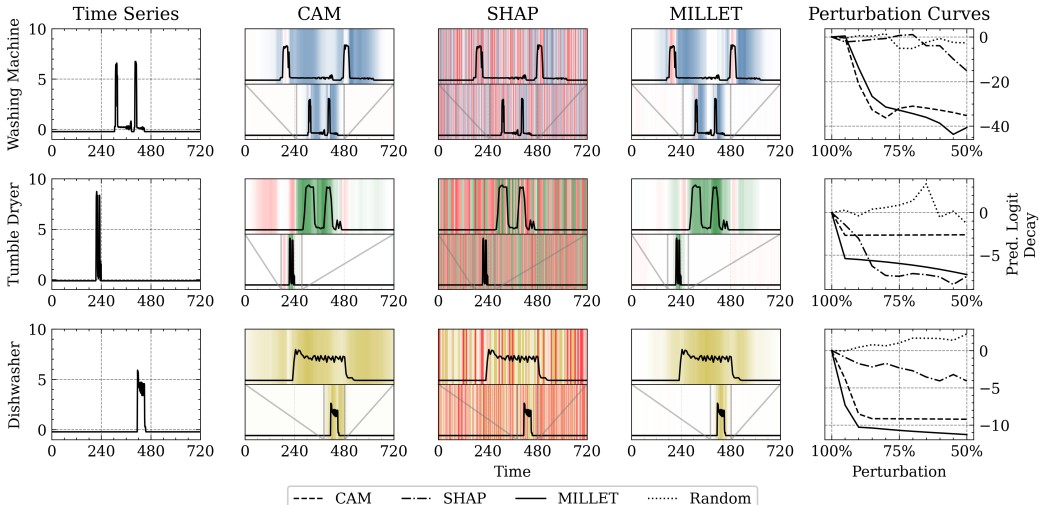

Figure 7: Comparison of interpretations on the `LargeKitchenAppliances` UCR dataset. **Left:** Original time series. **Middle:** Interpretability scores heatmap for CAM, SHAP, and `MILLET` as Fig. 1. **Right:** Perturbation curves showing the rate at which the model prediction decays when time points are removed following the orderings proposed by the different interpretability methods.

Fig. 7 shows interpretations for CAM, SHAP, and `MILLET` on `LargeKitchenAppliances`; a UCR dataset for identifying household electric appliances (Washing Machine, Tumble Dryer, or Dishwasher) from electricity usage. From the `MILLET` interpretability outputs, we identify that the model has learnt different motifs for each class: long periods of usage just above zero indicate Washing Machine, spikes above five indicate Tumble Dryer, and prolonged usage at just below five indicates Dishwasher. The Washing Machine example contains short spikes above five but `MILLET` identifies these as refuting the prediction, suggesting the model does not relate these spikes with the Washing Machine class. SHAP provides very noisy interpretations and does not show strong performance on the perturbation curves. Similar to our findings for *WebTraffic* (Sec. 4.2), `MILLET` provides sparser explanations than CAM, i.e. focusing on smaller regions and returning fewer discriminatory time points, which is helpful when explaining longer time series. We provide further analysis in the Appendix: head-to-heads (App. D.3), false negatives (App. E.1), dataset variance (App. E.2), model variance (App. E.3), run time (App. E.4), and an ablation study (App. E.5) .

## 6 CONCLUSION

The `MILLET` framework presented in this work is the first comprehensive analysis of MIL for TSC. Its positive value is demonstrated across the 85 UCR datasets and the *WebTraffic* dataset proposed in this work. `MILLET` provides inherent mechanisms to localise, interpret, and explain influences on model behaviour, and improves predictive accuracy in most cases. Through the transparent decision-making gained from `MILLET`, practitioners can improve their understanding of model dynamics without the need for expensive (and often ineffective) post-hoc explainability methods. In addition, `MILLET` explanations are sparse – they distill the salient signatures of classes to a small number of relevant sub-sequences, which is especially important for long time series. We believe this work lays firm foundations for increased development of MIL methods in TSC, and facilitates future work:

**Extension to more datasets:** This could include the full set of 142 datasets used in *Bake Off Redux* (Middlehurst et al., 2023), multivariate datasets, and variable length time series. Variable lengths would not require any methodological changes (contrary to several other methods), but multivariate settings would require interpretability to consider the input channel (Hsieh et al., 2021).

**Application to other models:** We have demonstrated the use of `MILLET` for DL TSC models. Future work could extend its use to other types of TSC models, e.g. the `ROCKET` (convolutional) family of methods (Dempster et al., 2020), which includes `Hydra-MR`. Our proposed pooling method, `Conjunctive`, is also applicable in general MIL problems beyond TSC.

**Pre-training/fine-tuning:** While we trained `MILLET` models in an end-to-end manner, an alternative approach is to take a pre-trained GAP model, replace the GAP layers with one of the proposed MIL pooling methods, and then fine-tune the network (facilitating faster training).

REPRODUCIBILITY STATEMENT

The code for this project was implemented in Python 3.8, with PyTorch as the main library for machine learning. A standalone code release is available at: `https://github.com/JAEarly/MILTimeSeriesClassification`. This includes our synthetic dataset and the ability to use our *plug-and-play* `MILLET` models.

Model training was performed using an NVIDIA Tesla V100 GPU with 16GB of VRAM and CUDA v12.0 to enable GPU support. For reproducibility, all experiments with a stochastic nature (e.g. model training, synthetic data generation, and sample selection) used pseudo-random fixed seeds. A list of all required libraries is given in the code release mentioned above.

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

# A    WHY MIL?

To answer the question of "why MIL", we first consider the requirements for interpreting TSC models. Our underlying assumption is that, for a classifier to predict a certain class for a time series, there must be one or more underlying motifs (a single time point or a collection of time points) in the time series that the model has identified as being indicative of the predicted class. In other words, only certain time points within a time series are considered discriminatory by the model (those that form the motifs), and the other time points are ignored (background time points or noise). This could also be extended to class refutation, where the model has learnt that certain motifs indicate that a time series does not belong to a particular class. To this end, the objective of TSC interpretability is then to uncover these supporting and refuting motifs, and present them as an explanation as to why a particular class prediction has been made.

We next consider how we could train a model to find these motifs if we already knew where and what they were, i.e. in a conventional supervised learning sense where the motifs are labelled. In this case, we would be able to train a model to predict a motif label from an input motif – in the simplest case this could be predicting whether a motif is discriminatory or non-discriminatory. This model could then be applied to an unlabelled time series and used to identify its motifs. Effectively, this hypothetical motif model is making time point level predictions.

Unfortunately, it is very difficult to train models in the above manner. The vast majority of time series datasets are only labelled at the time series level – no labels are provided at the time point level, therefore the motifs and their locations are unknown. While there are several successful deep learning models that are able to learn to make time series level predictions from time series level labels (e.g `FCN`, `ResNet`, `InceptionTime`), they are *black boxes* – they provide no inherent interpretation in the form of supporting/refuting motifs via time point level predictions. Without time point level labels, conventional supervised learning is unable to develop models that inherently make time point level predictions. While post-hoc interpretability methods could be used to uncover the motifs, they can be expensive (in the case of LIME and SHAP), or as we show in this work, inferior to inherent interpretability. As such, we can draw a set of requirements for a new and improved interpretable TSC approach:

1. **Inherent interpretability:** The model should provide time point predictions (i.e. motif identification) as part of its time series prediction process. This means interpretations are gained effectively for free as a natural byproduct of time series prediction. It also means expensive or ineffective post-hoc interpretability approaches are not required.

2. **Learn from time series labels:** As stated above, time series classification datasets rarely provide time point-level labels. Therefore, the model must be able to learn something insightful at the time point level given only time series labels.

3. **Provide a unified framework:** As there are a diverse range of existing TSC methods of different families, an effective TSC interpretability approach should be as widely applicable as possible to facilitate continued research in these different areas. This would also mean existing bodies of research can be applied in conjunction with the framework.

Given these requirements, we advocate for MIL as an appropriate method for an inherently interpretable TSC framework. MIL is designed for settings with bags of instances, where only the bags are labelled, not the instances. In a TSC setting, this equates to having time series labels without time point labels, which is exactly what we outlined above (and describes the vast majority of TSC datasets). Furthermore, certain types of existing MIL approaches, for example `Instance` and `Additive`, work by first making a prediction for every time point, and then aggregating over these predictions to make a time series prediction. This is inherently interpretable and facilitates motif identification. Finally, the over-arching concept of MIL for TSC, i.e. learning from time series labels but making both time point and time series predictions, is not tied to any one family of machine learning approach.

We also consider answers to potential questions about alternative solutions:

1. **Q: Why not label the motifs/time points to allow for supervised learning?**
   **A:** As discussed above, it is rare for a time series dataset to have time point labels. It could be possible to label the motifs, for example by having medical practitioners identify

discriminatory irregular patterns in ECG data. However, this labelling process is very time-consuming, and in some cases would require domain expertise, which is challenging and expensive to acquire. Furthermore, it would then require anyone interested in applying such supervised techniques to their own data to fully label everything at the time point level, increasing the cost and time of developing new datasets.

2. **Q: Why not label *some* of the motifs/time points and use a semi-supervised approach?**
**A:** While it might be possible to train a semi-supervised model that only requires some of the time points to be labelled, the model is no longer end-to-end. As the model only learns to predict at the time point level, it does not provide time series level predictions itself. Rather, some additional process is required to take the time point predictions and transform them into a time series prediction. Furthermore, a semi-supervised model has no method for leveraging both the time series labels and the partial time point labels.

3. **Q: What does MIL achieve beyond methods such as attention?**
**A:** While attention has been utilised previously in TSC, it does not provide class-specific interpretations, only general measures of importance across all classes. So while attention might identify motifs, it does not state which class these motifs belong to, nor whether they are supporting or refuting. Furthermore, attention is far less explicit than time point predictions – there is no guarantee that attention actually reflects the underlying motifs of decision-making, whereas in MIL the time point predictions directly determine the time series prediction.

## A.1    RELATED WORK (CONTINUED)

In this section, we further discuss existing TSC interpretability approaches and contrast them with our approach. MILLET is a time point-based explanation approach (following the taxonomy of Theissler et al., 2022), meaning it produces interpretations for each individual time point in a time series. This is in contrast to other types of methods such as subsequence-based techniques, which provide interpretations at a lower granularity. As such, we choose to compare against interpretability methods with the same level of granularity, i.e. those that also make time point-based interpretations (such as CAM and SHAP).

While alternative time series-specific SHAP approaches such as TimeSHAP (Bento et al., 2021) or WindowSHAP (Nayebi et al., 2023) could have been used, these both reduce the exponential sampling problem of SHAP by grouping time points together. As such, they do not have the same granularity as MILLET and do not meet our requirements. Two similar methods to SHAP are LIME (Ribeiro et al., 2016) and MILLI (Early et al., 2021), but these require careful tuning of hyperparameters (which SHAP does not), so are infeasible to run over the large number of datasets used in this work. LIMESegements (Sivill & Flach, 2022), an extension of LIME for TSC, has the same issue as TimeSHAP and WindowSHAP mentioned above: it groups time points into subsequences so does not produce interpretations at the same granularity as MILLET. Alternative approaches for TSC interpretability have similar granularity issues or are often very expensive to compute (relative to the cost of using MILLET or CAM). Such methods include Dynamask (Crabbé & Van Der Schaar, 2021), WinIT (Leung et al., 2022), and TimeX (Queen et al., 2023). If MILLET were to be extended to multivariate TSC problems, it would be compatible with Temporal Saliency Rescaling (TSR; Ismail et al., 2020), which has been shown to improve time point-based interpretability methods in multivariate settings.

While our choice of interpretability evaluation metrics are based on those used by Early et al. (2021), different metrics are used in existing TSC interpretability works, but these come with their own set of disadvantages. Metrics such as area under the precision curve (AUP) and area under the recall curve (AUR) are useful in that they separate whether the identified time points are indeed discriminatory (precision) from whether all discriminatory time points are identified (recall/coverage). While this would be beneficial in evaluating the effect of MILLET sparsity, it does not account for the ordering of time points in the interpretation, i.e. we want to reward the interpretation method for placing discriminatory time points earlier in the ordering (and punish it for placing non-discriminatory time points earlier on); this is something that NDCG@n achieves. The mean rank metric (Leung et al., 2022) also suffers from this issue — it is effectively an unweighted version of NDCG@n. Furthermore, AUP and AUR metrics require time point labels and sometimes need access to the underlying data generation process for resampling; AOPCR does not need either.

## B  MODEL DETAILS

### B.1  MILLET MODEL ARCHITECTURES

In the following section, we provide details on our `MILLET` models. For conciseness, we omit details on the backbone architectures. See Wang et al. (2017) for details on `FCN` and `ResNet`, and Ismail Fawaz et al. (2020) for details on `InceptionTime`. Each of the three feature extractor backbones used in this work produce fixed-size time point embeddings of length 128: $\mathbf{Z_i} \in \mathbb{R}^{t \times 128} = [\mathbf{z_i^1}, \mathbf{z_i^2}, \dots, \mathbf{z_i^t}]$, where $t$ is the input time series length. This is the initial *Feature Extraction* phase and uses unchanged versions of the original backbones.

A breakdown of the general model structure is given in Eqn. A.1. Note how *Positional Encoding* is only applied after *Feature Extraction*. Furthermore, *Positional Encoding* and *Dropout* are only applied in `MILLET` models, i.e. when using `Instance`, `Attention`, `Additive`, or `Conjunctive` pooling.

$$\text{Feature Extraction} \rightarrow \text{Positional Encoding} \rightarrow \text{Dropout} \rightarrow \text{MIL Pooling} \quad\quad \text{(A.1)}$$

Below we detail the *Positional Encoding* processes, and then give architectures for each of the *MIL Pooling* approaches. These are kept unchanged across backbones.

### B.1.1  POSITIONAL ENCODING

Our approach for *Positional Encoding* uses fixed positional encodings (Vaswani et al., 2017):

$$PE_{(pos, 2i)} = \sin(pos/10000^{2i/d_{model}}),$$
$$PE_{(pos, 2i+1)} = \cos(pos/10000^{2i/d_{model}}), \quad\quad \text{(A.2)}$$

where $pos = [1, \dots, t]$ is the position in the time series, $d_{model} = 128$ is the size of the time point embeddings, and $i = [1, \dots, d_{model}/2]$. As such, the *Positional Encoding* output is the same shape as the input time point embeddings ($\mathbb{R}^{t \times 128}$). The positional encodings are then simply added to the time point embeddings.

In cases where time points are removed from the bag, e.g. when calculating AOPCR (see App. D.1), we ensure the positional encodings remain the same for the time points that are still in the bag. For example, if the first 20 time points are removed from a time series, the 21st time point will still have positional encoding $PE_{(21)}$, not $PE_{(1)}$.

### B.1.2  MIL POOLING ARCHITECTURES

In Tables A.1 to A.5 we provide architectures for the different MIL pooling methods used in this work (see Fig. 2 for illustrations). Each row describes a layer in the pooling architecture. In each case, the input is a bag of time point embeddings (potentially with positional encodings and dropout already applied, which does not change the input shape; see App. B.1.1). The input is batched with a batch size of $b$, and each time series is assumed to have the same length $t$. Therefore, the input is four dimensional: batch size × number of channels × time series length × embedding size. However, in this work, as we are using univariate time series, the number of channels is always one. The problem has $c$ classes, and the pooling methods produce logit outputs – softmax is later applied as necessary.

Table A.1: MIL Pooling: `Embedding` (GAP).

| Process | Layer | Input | Output |
|---|---|---|---|
| Pooling | Mean | $b \times 1 \times t \times 128$ (Time Point Embs.) | $b \times 1 \times 1 \times 128$ (TS Emb.) |
| Classifier | Linear | $b \times 1 \times 1 \times 128$ (Time Series Emb.) | $b \times 1 \times 1 \times c$ (TS Pred.) |

Table A.2: MIL Pooling: `Attention`. We use an internal dimension of 8 in the attention head, and apply sigmoid rather than softmax due to the possibility of long time series. Attention weighting scales the time point embeddings by their respective attention scores.

| Process | Layer | Input | Output |
|---|---|---|---|
| Attention | Linear + tanh | $b \times 1 \times t \times 128$ (Time Point Embs.) | $b \times 1 \times t \times 8$ |
| | Linear + sigmoid | $b \times 1 \times t \times 8$ | $b \times 1 \times t \times 1$ (Attn. Scores) |
| Pooling | Attn. Weighting | $b \times 1 \times t \times 128$ (Time Point Embs.) | $b \times 1 \times t \times 128$ |
| | Mean | $b \times 1 \times t \times 128$ | $b \times 1 \times 1 \times 128$ (TS Emb.) |
| Classifier | Linear | $b \times 1 \times 1 \times 128$ (Time Series Emb.) | $b \times 1 \times 1 \times c$ (TS Pred.) |

Table A.3: MIL Pooling: `Instance`.

| Process | Layer | Input | Output |
|---|---|---|---|
| Classifier | Linear | $b \times 1 \times t \times 128$ (Time Point Embs.) | $b \times 1 \times t \times c$ (TP Preds.) |
| Pooling | Mean | $b \times 1 \times t \times c$ (Time Point Preds.) | $b \times 1 \times 1 \times c$ (TS Pred.) |

Table A.4: MIL Pooling: `Additive`.

| Process | Layer | Input | Output |
|---|---|---|---|
| Attention | Linear + tanh | $b \times 1 \times t \times 128$ (Time Point Embs.) | $b \times 1 \times t \times 8$ |
| | Linear + sigmoid | $b \times 1 \times t \times 8$ | $b \times 1 \times t \times 1$ (Attn. Scores) |
| Classifier | Attn. Weighting | $b \times 1 \times t \times 128$ (Time Point Embs.) | $b \times 1 \times t \times 128$ |
| | Linear | $b \times 1 \times t \times 128$ | $b \times 1 \times t \times c$ (TP Preds.) |
| Pooling | Mean | $b \times 1 \times t \times c$ (Time Point Preds.) | $b \times 1 \times 1 \times c$ (TS Pred.) |

Table A.5: MIL Pooling: `Conjunctive`. In this case, attention weighting scales the time point predictions by their respective attention scores, rather than scaling the embeddings.

| Process | Layer | Input | Output |
|---|---|---|---|
| Attention | Linear + tanh | $b \times 1 \times t \times 128$ (Time Point Embs.) | $b \times 1 \times t \times 8$ |
| | Linear + sigmoid | $b \times 1 \times t \times 8$ | $b \times 1 \times t \times 1$ (Attn. Scores) |
| Classifier | Linear | $b \times 1 \times t \times 128$ (Time Point Embs.) | $b \times 1 \times t \times c$ (TP Preds.) |
| Pooling | Attn. Weighting | $b \times 1 \times t \times c$ (Time Point Preds.) | $b \times 1 \times t \times c$ |
| | Mean | $b \times 1 \times t \times c$ | $b \times 1 \times 1 \times c$ (TS Pred.) |

## B.2 TRAINING AND HYPERPARAMETERS

In this work, all models were trained in the same manner. We used the Adam optimiser with a fixed learning rate of 0.001 for 1500 epochs, and trained to minimise cross entropy loss. Training was performed in an end-to-end manner, i.e. all parts of the networks (including the backbone feature extraction layers) were trained together, and no pre-training or fine-tuning was used. Dropout (if used) was set to 0.1, and batch size was set to $\min(16, \lfloor num\ training\ time\ series/10 \rfloor)$ to account for datasets with small training set sizes. For example, if a dataset contains only 100 training time series, the batch size is set to 10.

No tuning of hyperparameters was used – values were set based on the those used for training the original backbone models. As the existing DL TSC methods used fixed hyperparameters, our decision not to tune the hyperparameters facilitates a fairer comparison. It also has the benefit of providing a robust set of default values for use in derivative works. However, better performance could be achieved by tuning hyperparameters for individual datasets. We would expect hyperparameter tuning for `MILLET` to have a greater impact than doing so for the GAP versions of the models as

`MILLET` provides more scope for tuning (e.g. dropout, attention-head size, and whether to include positional encodings). As greater flexibility is achieved by having the ability to add/remove/tune the additional elements of `MILLET`, this would lead to more specialised models (and larger performance improvements) for each dataset. For example, in our ablation study (App. E.5), we found that positional encoding was beneficial for some datasets but not others.

No validation datasets were used during training or evaluation . Instead, the final model weights were selected based on the epoch that provides the lowest training loss. As such, training was terminated early if a loss of zero was reached (which was a very rare occurrence, but did happen). Models started with random weight initialisations, but pseudo-random fixed seeds were used to enable reproducibility. For repeat training, the seeds were different for each repeat (i.e. starting from different random initialisations), but these were consistent across models and datasets.

### B.3  SHAP Details

In our SHAP implementation we used random sampling of coalitions. Guided sampling (selecting coalitions to maximise the SHAP kernel) would have proved too expensive: the first coalitions sampled would be all the single time point coalitions and all the $t - 1$ length coalitions (for a time series of length $t$), which results in $2t$ coalitions and thus $2t$ calls to the model. In the case of *WebTraffic*, this would be 2016 samples, rather than the 500 we used with random sampling (which still took far longer to run than `MILLET`, see App. E.4). Furthermore, Early et al. (2021) showed random sampling to be equal to or better than guided sampling in some cases.

## C  Dataset Details

### C.1  Synthetic Dataset Details (*WebTraffic*)

In our synthetic time series dataset, *WebTraffic*, each time series is a week long with a sample rate of 10 minutes ($60 * 24 * 7/10 = 1008$ time points). The training and test set are independently generated using fixed seeds to facilitate reproducibility, and are both balanced with 50 time series per class (500 total time series for each dataset).

To explain the synthetic time series generation process, we first introduce a new function:

$$WarpedSin(a, b, p, s, x) = \frac{a}{2} \sin\left( x' - \frac{\sin(x')}{s} \right) + b, \text{where } x' = 2\pi(x - p). \quad \text{(A.3)}$$

Parameters $a$, $b$, $p$, and $s$ control amplitude, bias (intercept), phase, and skew respectively. $WarpedSin$ is used to generate daily seasonality in the following way:

$$SampleDay(a_D, b, p, s, \sigma, j) = \mathcal{N}(RateDay(a_D, b, p, s, j), \sigma), \quad \text{(A.4)}$$

$$RateDay(a_D, b, p, s, j) = WarpedSin(a_D, b, p + 0.55, s, j/144) \quad \text{(A.5)}$$

where $j \in [1, \ldots, 1008]$ is the time index and $\sigma$ is a parameter that controls the amount of noise created when sampling (via a normal distribution). The daily rates provide daily seasonality (i.e. peaks in the evening and troughs in the morning). However, to take this further we also add weekly seasonality (i.e. more traffic at the weekends than early in the week). To do so, we further utilise $WarpedSin$:

$$RateWeek(a_W, j) = WarpedSin(a_W, 1, 0.6, 2, j/1008). \quad \text{(A.6)}$$

To add this weekly seasonality, we multiply the daily sampled values by the weekly rate. Therefore, to produce a base time series, we arrive at a formula with six parameters:

$$SampleWeek(a_D, a_W, b, p, s, \sigma) = SampleDay(a_D, b, p, s, \sigma, j) * RateWeek(a_W, j)$$
$$\text{for } j \in [1, \dots, 1008]. \quad \text{(A.7)}$$

To generate a collection of $n$ time series, we sample the following parameter distributions $n$ times (i.e. once for every time series we want to generate):

- Amplitude daily: $a_D \sim \mathcal{U}_\mathbb{R}(2, 4)$

- Amplitude weekly $a_W \sim \mathcal{U}_\mathbb{R}(0.8, 1.2)$

- Bias $b \sim \mathcal{U}_\mathbb{R}(2.5, 5)$

- Phase $p \sim \mathcal{U}_\mathbb{R}(-0.05, 0.05)$

- Skew $s \sim \mathcal{U}_\mathbb{R}(1, 3)$

- Noise $\sigma \sim \mathcal{U}_\mathbb{R}(2, 4)$

We use $a \sim \mathcal{U}_\mathbb{R}(b, c)$ to denote uniform random sampling between $b$ and $c$, where $a \in \mathbb{R}$. Below, we also use uniform random *integer* sampling $a' \sim \mathcal{U}_\mathbb{Z}(b, c)$, where $a' \in \mathbb{Z}$.

The above generation process results in a collection of $n$ time series, but currently they are all class zero (Class 0: None) as they have had no signatures injected. We describe how we inject each of the nine signature types below. Aside from the Spikes signature (Class 1), all signatures are injected in random windows of length $l \sim \mathcal{U}_\mathbb{Z}(36, 288)$ starting in position $p \sim \mathcal{U}_\mathbb{Z}(0, t - l)$. The minimum window size of 36 corresponds to 0.25 days, and the maximum size of 288 corresponds to 2 days. In all cases, values are clipped to be non-negative, i.e. all time points following signature injection are $\geq 0$. These methods are inspired by, but not identical to, the work of Goswami et al. (2023). We provide an overview of the entire synthetic dataset generation process in Fig. A.1. Exact details on the injected signatures are given below, along with focused examples in Fig. A.2.

**Class 1: Spikes** Spikes are injected at random time points throughout the time series, with probability $p = 0.01$ for each time point. The magnitude of a spike is drawn from $\mathcal{N}(3.0, 2.0)$, and then added to or subtracted from the original time point value with equal probability.

**Class 2: Flip** The randomly selected window is flipped in the time dimension.

**Class 3: Skew** A skew is applied to the time points in the randomly selected window. A skew amount is first sampled from $\mathcal{U}_\mathbb{R}(0.25, 0.45)$, which is then add to or subtracted from 0.5 with equal probability. This gives a new skew value $w \in [0.05, 0.25] \cup [0.75, 0.95]$. The random window is then interpolated such that the value at the midpoint is now located at time point $\lfloor w * l \rfloor$ within the window, i.e. stretching the time series on one side and compressing it on the other.

**Class 4: Noise** Noise is added to the random window. The amount of noise is first sampled from $\sigma_{Noise} \sim \mathcal{U}_\mathbb{R}(0.5, 1.0)$. Then, for each time point in the selected window, noise is added according to $\mathcal{N}(0, \sigma_{Noise})$.

**Class 5: Cutoff** A cutoff value is sampled from $c \sim \mathcal{U}_\mathbb{R}(0.0, 0.2)$ The values in the random window are then set to $\mathcal{N}(c, 0.1)$.

**Class 6: Average** This signature is the opposite of noise injection, i.e. applying smoothing to the values in the random window. This is achieved through applying a moving average with a window size sampled from $\mathcal{U}_\mathbb{Z}(5, 10)$.

**Class 7: Wander** A linear trend is applied to values in the random window. The trend linearly transitions from 0 to $\mathcal{U}_\mathbb{R}(2.0, 3.0)$, and is then added to or subtracted from the values in the window with equal probability.

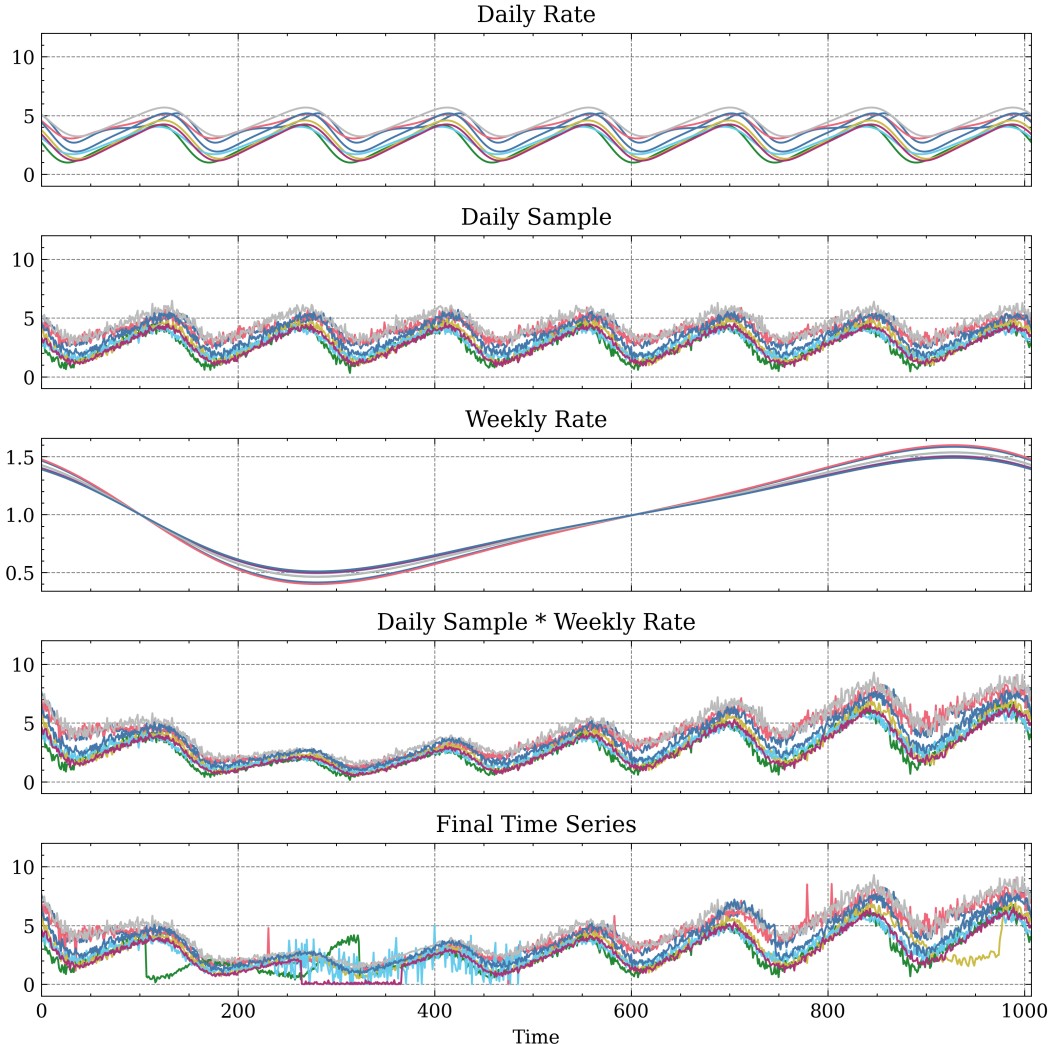

Figure A.1: An overview of our synthetic dataset generation. From top to bottom: daily seasonality rate, daily seasonality with sampled noise, weekly seasonality rate, base time series with daily and weakly seasonality, and final time series with signatures injected into the base time series.

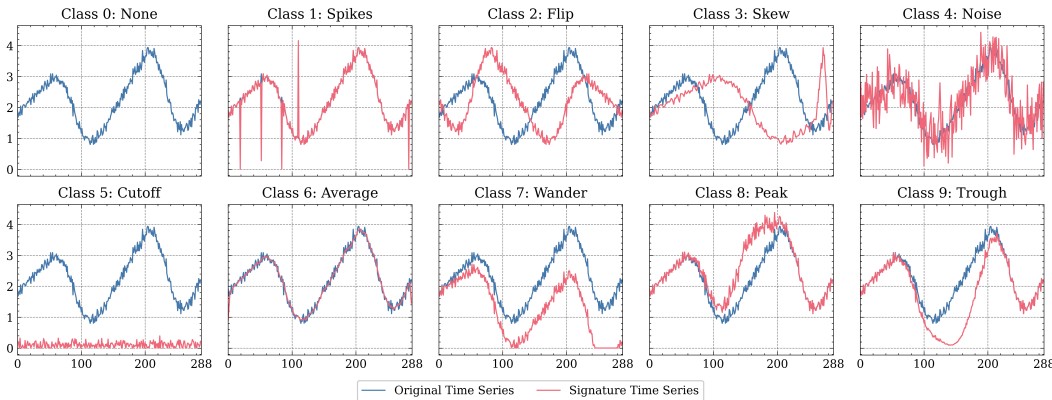

Figure A.2: Examples of injected signatures. Each example focuses on the window in which the signatures are injected (i.e. omitting the rest of the time series), and windows are set to a fixed length of 288 (the maximum length when selecting random windows) to aid visualisation.

**Class 8: Peak**   A smooth peak is created from the probability density function (PDF) of $\mathcal{N}(0, 1)$ from -5 to 5, and then the values are multiplied by a scalar sampled from $\mathcal{U}_{\mathbb{R}}(1.5, 2.5)$. Values in the random window are then multiplied by the values of the peak, creating a smooth transition from the existing time series.

**Class 9: Trough**   The same method to generate the Peak signatures is used to generate a trough, but the PDF values are instead multiplied by $\mathcal{U}_{\mathbb{R}}(-2.5, -1.5)$ (same scalar sample range but negative).

## C.2   UCR DATASET DETAILS

For the UCR datasets, we used the original train/test splits as provided from the archive source.[8] $z$-normalisation was applied to datasets that were not already normalised. The exhaustive list of univariate UCR datasets used in this work is:

```
Adiac, ArrowHead, Beef, BeetleFly, BirdChicken, Car, CBF,
ChlorineConcentration, CinCECGTorso, Coffee, Computers,
CricketX, CricketY, CricketZ, DiatomSizeReduction,
DistalPhalanxOutlineAgeGroup, DistalPhalanxOutlineCorrect,
DistalPhalanxTW, Earthquakes, ECG200, ECG5000, ECGFiveDays,
ElectricDevices, FaceAll, FaceFour, FacesUCR, FiftyWords,
Fish, FordA, FordB, GunPoint, Ham, HandOutlines, Haptics,
Herring, InlineSkate, InsectWingbeatSound, ItalyPowerDemand,
LargeKitchenAppliances, Lightning2, Lightning7, Mallat,
Meat, MedicalImages, MiddlePhalanxOutlineAgeGroup,
MiddlePhalanxOutlineCorrect, MiddlePhalanxTW, MoteStrain,
NonInvasiveFetalECGThorax1, NonInvasiveFetalECGThorax2,
OliveOil, OSULeaf, PhalangesOutlinesCorrect, Phoneme, Plane,
ProximalPhalanxOutlineAgeGroup, ProximalPhalanxOutlineCorrect,
ProximalPhalanxTW, RefrigerationDevices, ScreenType, ShapeletSim,
ShapesAll, SmallKitchenAppliances, SonyAIBORobotSurface1,
SonyAIBORobotSurface2, StarLightCurves, Strawberry, SwedishLeaf,
Symbols, SyntheticControl, ToeSegmentation1, ToeSegmentation2,
Trace, TwoLeadECG, TwoPatterns, UWaveGestureLibraryAll,
UWaveGestureLibraryX, UWaveGestureLibraryY, UWaveGestureLibraryZ,
Wafer, Wine, WordSynonyms, Worms, WormsTwoClass, Yoga.
```

# D   ADDITIONAL RESULTS

## D.1   INTERPRETABILITY METRICS

Below we provide more details on the metrics used to evaluate interpretability, which are based on the process proposed by Early et al. (2021).

**AOPCR: Evaluation without time point labels**   When time point labels are not present, the model can be evaluated via perturbation analysis. The underlying intuition is that, given a correct ordering of time point importance in a time series, iteratively removing the most important (discriminatory) time points should cause the model prediction to rapidly decrease. Conversely, a random or incorrect ordering will lead to a much slower decrease in prediction. Formally, when evaluating the interpretations generated by a classifier $F_c$ for a time series $\mathbf{X_i} = \{\mathbf{x_i^1}, \mathbf{x_i^2}, \ldots, \mathbf{x_i^t}\}$ with respect to class $c$, we first re-order the time series according to the importance scores (with the most important time points first): $\mathbf{O_{i,c}} = \{\mathbf{o_i^1}, \mathbf{o_i^2}, \ldots, \mathbf{o_i^t}\}$. The perturbation metric is then calculated by:

$$AOPC(\mathbf{X_i}, \mathbf{O_{i,c}}) = \frac{1}{t-1} \sum_{j=1}^{t-1} F_c(\mathbf{X_i}) - F_c(MoRF(\mathbf{X_i}, \mathbf{O_{i,c}}, j)), \tag{A.8}$$

$$\text{where } MoRF(\mathbf{X_i}, \mathbf{O_{i,c}}, j) = MoRF(\mathbf{X_i}, \mathbf{O_{i,c}}, j-1) \setminus \{\mathbf{o_i^j}\},$$
$$\text{and } MoRF(\mathbf{X_i}, \mathbf{O_{i,c}}, 0) = \mathbf{X_i}.$$

---

[8] https://www.cs.ucr.edu/~eamonn/time_series_data_2018/

$MoRF$ is used to signify the ordering is Most Relevant First. In Eqn. A.8, the perturbation curve is calculated by removing individual time points and continues until all but one time point (the least important as assessed by the model) is left. This is expensive to compute, as a call to the model must be made for each perturbation. To improve the efficiency of this calculation, we group time points together into blocks equal to 5% of the total time series length, and only perturb the time series until 50% of the time points have been removed. As such, we only need to make 10 calls to the model per time series evaluation.

To facilitate better comparison between models, we normalise by comparing to a random ordering. To compensate for the stochastic nature of using random orderings, we average over three different random orderings, where $\mathbf{R_i^{(r)}}$ is the $r^{th}$ repeat random ordering:

$$AOPCR(\mathbf{X_i}, \mathbf{O_{i,c}}) = \frac{1}{3} \sum_{r=1}^{3} \left( AOPC\left(\mathbf{X_i}, \mathbf{O_{i,c}}\right) - AOPC(\mathbf{X_i}, \mathbf{R_i^{(r)}}) \right). \tag{A.9}$$

**NDCG@n: Evaluation with time point labels** If the time point labels are known, a perfect ordering of time point importance would have every discriminatory time point occurring at the start. If there are $n$ discriminatory time points, we would expect to see these in the first $n$ places in the ordered interpretability output. The fewer true discriminatory time points there are in the first $n$ places, the worse the interpretability output. Furthermore, we want to reward the model for placing discriminatory time points earlier in the ordering (and punish it for placing non-discriminatory time points earlier on). This can be achieved by placing a higher weight on the start of the ordering. Formally,

$$NDCG@n(\mathbf{O_{i,c}}) = \frac{1}{IDCG} \sum_{j=1}^{n} \frac{rel(\mathbf{O_{i,c}}, j)}{log_2(j+1)}, \tag{A.10}$$

$$\text{where } IDCG = \sum_{j=1}^{n} \frac{1}{log_2(j+1)},$$

$$\text{and } rel(\mathbf{O_{i,c}}, j) = \begin{cases} 1 \text{ if } \mathbf{o_i^j} \text{ is a discriminatory time point,} \\ 0 \text{ otherwise.} \end{cases}$$

## D.2 *WebTraffic* ADDITIONAL RESULTS

We first provide a complete set of results for predictive performance on *WebTraffic*, comparing the GAP with `MILLET`. Tables A.6, A.7, A.8 give results on accuracy, AUROC, and loss respectively.



Table A.6: *WebTraffic* Accuracy.

|  | FCN | ResNet | ITime | Mean |
|---|---|---|---|---|
| GAP | 0.756 | 0.860 | 0.934 | 0.850 |
| Attention | **0.820** | **0.866** | 0.936 | **0.874** |
| Instance | 0.782 | 0.862 | 0.938 | 0.861 |
| Additive | 0.814 | 0.858 | **0.940** | 0.871 |
| Conjunctive | 0.818 | 0.850 | **0.940** | 0.869 |

Table A.7: *WebTraffic* AUROC.

|  | FCN | ResNet | ITime | Mean |
|---|---|---|---|---|
| GAP | 0.961 | 0.982 | **0.997** | 0.980 |
| Attention | **0.973** | **0.984** | **0.997** | 0.984 |
| Instance | 0.962 | 0.982 | **0.997** | 0.980 |
| Additive | **0.973** | **0.984** | **0.997** | 0.984 |
| Conjunctive | **0.973** | **0.984** | 0.996 | **0.985** |



Table A.8: *WebTraffic* Loss.

|  | FCN | ResNet | ITime | Mean |
|---|---|---|---|---|
| GAP | 0.939 | **0.633** | 0.268 | 0.614 |
| Attention | **0.863** | 0.701 | 0.279 | 0.614 |
| Instance | 0.871 | 0.709 | 0.257 | 0.612 |
| Additive | 0.882 | 0.678 | **0.252** | 0.604 |
| Conjunctive | 0.866 | 0.638 | 0.277 | **0.594** |

Table A.9 shows the complete interpretability results. Best performance in each case comes from one of `MILLET Instance`, `Additive`, or `Conjunctive`. The exception is NDCG@n for CAM on `InceptionTime`, which, as discussed in Section 4.2, is likely due to the sparsity of `MILLET` explanations. We also note that SHAP performs very poorly across all pooling methods, and that `Attention` is also worse than the other methods (as it does not make class-specific explanations).

Table A.9: Interpretability performance (AOPCR / NDCG@n) on our *WebTraffic* dataset. Results are generated using the ensembled versions of the models. This is an expanded version of Table 1.

| | FCN | ResNet | InceptionTime | Mean |
|---|---|---|---|---|
| CAM | 12.780 / 0.532 | 20.995 / 0.582 | 12.470 / **0.707** | 15.415 / 0.607 |
| SHAP - `Attention` | 1.507 / 0.271 | -3.293 / 0.250 | -5.376 / 0.249 | -2.387 / 0.257 |
| SHAP - `Instance` | 1.977 / 0.283 | -0.035 / 0.257 | -4.020 / 0.259 | -0.692 / 0.266 |
| SHAP - `Additive` | 0.900 / 0.270 | -1.077 / 0.250 | -5.952 / 0.249 | -2.043 / 0.256 |
| SHAP - `Conjunctive` | 0.987 / 0.267 | -0.552 / 0.257 | -5.359 / 0.246 | -1.641 / 0.257 |
| SHAP Best | 1.977 / 0.283 | -0.035 / 0.257 | -4.020 / 0.259 | -0.692 / 0.266 |
| `MILLET` - `Attention` | 4.780 / 0.425 | 6.382 / 0.380 | -1.597 / 0.420 | 3.188 / 0.408 |
| `MILLET` - `Instance` | 12.841 / **0.540** | 23.090 / 0.584 | **13.192** / 0.704 | 16.374 / **0.609** |
| `MILLET` - `Additive` | **14.522** / 0.532 | **24.880** / 0.589 | 10.274 / 0.684 | **16.559** / 0.602 |
| `MILLET` - `Conjunctive` | 13.221 / 0.539 | 24.597 / **0.591** | 11.100 / 0.694 | 16.306 / 0.608 |
| `MILLET Best` | **14.522** / **0.540** | **24.880** / **0.591** | **13.192** / 0.704 | **17.531** / **0.612** |

### D.3 UCR ADDITIONAL RESULTS

In this section we give extended results on the UCR datasets. First, Table A.10 compares predictive performance of the `MILLET` methods – we find that `Conjunctive` gives the best average performance. In Table A.11 we give `MILLET` interpretability results on the UCR datasets for both individual and ensemble models. Interestingly, we find that `MILLET` performs well on all cases except the ensemble `ResNet` models. In this case, its performance drops significantly compared to the individual ResNet performance – something that is not observed for the other backbones. We observe something similar in our ablation study, see App. E.5. We then compare `MILLET` performance with that of six SOTA methods in Table A.12.[9]

Table A.10: `MILLET` predictive performance (accuracy / balanced accuracy) on 85 UCR datasets.

| | FCN | ResNet | InceptionTime | Mean |
|---|---|---|---|---|
| GAP | 0.828 / 0.804 | 0.843 / 0.819 | 0.853 / 0.832 | 0.841 / 0.818 |
| `Attention` | 0.781 / 0.754 | **0.846** / **0.823** | 0.855 / 0.832 | 0.827 / 0.803 |
| `Instance` | 0.829 / 0.804 | 0.842 / 0.818 | 0.855 / 0.833 | 0.842 / 0.819 |
| `Additive` | 0.835 / 0.810 | 0.845 / 0.822 | 0.855 / 0.832 | 0.845 / 0.822 |
| `Conjunctive` | **0.838** / **0.814** | 0.845 / 0.822 | **0.856** / **0.834** | **0.846** / **0.823** |

Table A.11: `MILLET` AOPCR interpretability performance (individual / ensemble) on 85 UCR datasets. Best results over all `MILLET` models is given for reference.

| | FCN | ResNet | InceptionTime | Mean |
|---|---|---|---|---|
| GAP | 6.518 / 6.534 | 6.341 / **6.445** | 3.392 / 4.144 | 5.417 / 5.707 |
| `Attention` | -0.023 / 0.474 | 0.620 / 1.513 | -0.909 / -0.936 | -0.104 / 0.351 |
| `Instance` | **6.868** / **6.925** | 6.338 / 5.903 | 4.260 / 3.973 | 5.822 / 5.600 |
| `Additive` | 6.443 / 6.298 | **6.526** / 5.871 | **4.963** / **5.083** | **5.977** / 5.751 |
| `Conjunctive` | 6.361 / 6.498 | 6.438 / 6.006 | 4.553 / 4.764 | 5.784 / **5.756** |
| *MILLET Best* | *6.868 / 6.925* | *6.526 / 6.006* | *4.963 / 5.083* | *6.119 / 6.004* |

---

[9]SOTA results obtained from *Bake Off Redux* using column zero of the results files (original train/test split). We did not use the `FCN`, `ResNet`, or `ITime` results as we trained our own versions of these models.

Table A.12: Results for `MILLET` against baselines on 85 UCR datasets. Results are given in the form mean / rank / number of wins. `HC2` is included for reference but not directly compared to.

| Method | Accuracy ↑ | Bal. Accuracy ↑ | AUROC ↑ | NLL ↓ |
|---|---|---|---|---|
| Hydra-MR | **0.857** / **8.688** / **19** | 0.831 / 9.994 / **17** | 0.875 / 18.647 / 7 | 0.953 / 11.194 / 9 |
| FreshPRINCE | 0.833 / 12.841 / 14 | 0.801 / 13.824 / 13 | **0.944** / 10.206 / 19 | **0.714** / 10.800 / **11** |
| PF | 0.821 / 14.729 / 8 | 0.795 / 15.000 / 6 | 0.924 / 12.141 / 14 | 0.709 / 10.906 / 4 |
| RDST | 0.850 / 10.729 / 13 | 0.822 / 11.529 / 11 | 0.870 / 19.165 / 6 | 0.997 / 12.612 / 9 |
| RSTSF | 0.842 / 12.382 / 12 | 0.810 / 13.835 / 10 | 0.945 / 9.835 / 22 | 0.723 / 11.212 / 8 |
| WEASEL-D | 0.850 / 10.376 / 17 | 0.823 / 11.535 / 13 | 0.871 / 19.559 / 6 | 0.999 / 12.288 / 7 |
| FCN | 0.828 / 15.053 / 6 | 0.804 / 14.512 / 6 | 0.929 / 13.818 / 13 | 1.038 / 11.529 / 3 |
| Attn. FCN | 0.781 / 14.929 / 6 | 0.754 / 14.547 / 5 | 0.927 / 13.029 / 12 | 1.194 / 13.082 / 2 |
| Ins. FCN | 0.829 / 14.653 / 7 | 0.804 / 14.376 / 8 | 0.930 / 13.353 / 18 | 1.023 / 10.412 / 3 |
| Add. FCN | 0.835 / 13.547 / 5 | 0.810 / 13.100 / 5 | 0.933 / 11.594 / 14 | 0.994 / 10.247 / 3 |
| Conj. FCN | 0.838 / 12.624 / 6 | 0.814 / 12.247 / 6 | 0.934 / 11.012 / 16 | 0.973 / **9.635** / 3 |
| ResNet | 0.843 / 11.741 / 7 | 0.819 / 11.271 / 6 | 0.937 / 10.559 / 18 | 1.091 / 13.024 / 0 |
| Attn. ResNet | 0.846 / 11.400 / 8 | 0.823 / 10.671 / 9 | 0.939 / 9.888 / 13 | 1.051 / 12.188 / 1 |
| Ins. ResNet | 0.842 / 11.918 / 10 | 0.818 / 11.653 / 9 | 0.936 / 10.394 / 17 | 1.071 / 12.553 / 2 |
| Add. ResNet | 0.845 / 11.147 / 10 | 0.822 / 10.500 / 9 | 0.936 / 10.282 / 14 | 1.073 / 13.082 / 0 |
| Conj. ResNet | 0.845 / 11.335 / 10 | 0.822 / 10.806 / 10 | 0.939 / 9.329 / 15 | 1.035 / 12.176 / 1 |
| ITime | 0.853 / 9.724 / 16 | 0.832 / 8.965 / 15 | 0.939 / 9.500 / 24 | 1.078 / 11.571 / 7 |
| Attn. ITime | 0.855 / 9.512 / 10 | 0.832 / 9.018 / 11 | 0.940 / 8.876 / 20 | 1.069 / 11.618 / 3 |
| Ins. ITime | 0.855 / 9.471 / 16 | 0.833 / 9.053 / 16 | 0.940 / 8.406 / **25** | 1.050 / 10.929 / 5 |
| Add. ITime | 0.855 / 9.235 / 11 | 0.832 / 8.729 / 12 | 0.940 / **8.394** / 21 | 1.067 / 11.488 / 3 |
| Conj. ITime | 0.856 / 8.976 / 15 | **0.834** / **8.482** / **17** | 0.939 / 9.006 / 22 | 1.085 / 12.065 / 4 |
| *HC2* | *0.860 / 7.988 / 20* | *0.830 / 9.353 / 17* | *0.950 / 6.006 / 42* | *0.607 / 8.388 / 14* |

We perform a further direct comparison against the best two SOTA results, `HC2` and `Hydra-MR`, allowing us to evaluate how many and on which datasets `MILLET` performs better. As shown in Fig. A.3, we find that our best-performing `MILLET` approach (`Conjunctive InceptionTime`) wins or draws on 48/85 (56.5%), 49/85 (57.7%), and 52/85 (61.2%) UCR datasets against `InceptionTime`, `HC2`, and `Hydra-MR` respectively.

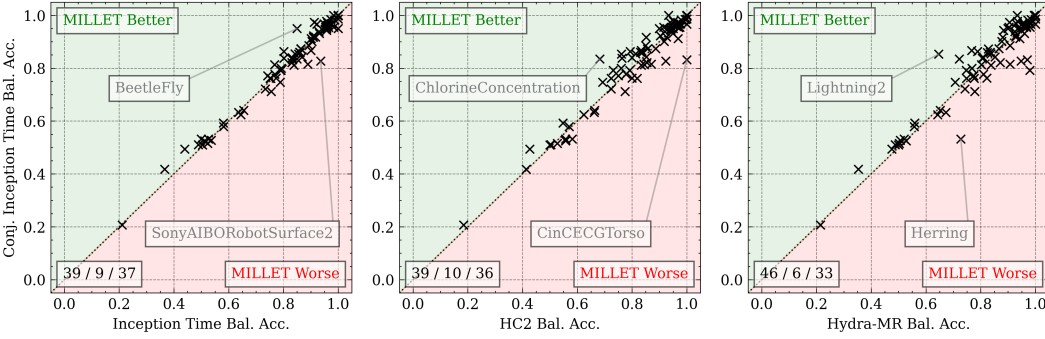

Figure A.3: Direct comparison of our best `MILLET` method against SOTA. Numbers in the bottom left indicate the number of wins / draws / losses for `Conjunctive InceptionTime`. Datasets with the greatest positive and negative differences are indicated.

## E ADDITIONAL EXPERIMENTS

### E.1 INVESTIGATION OF FALSE NEGATIVES

Our experiments showed `Conjunctive` pooling gives better predictive and interpretability performance than other pooling methods. To further investigate why this is the case, we analysed its performance on our proposed *WebTraffic* dataset. We found its accuracy was between 0.86 and 1.0 for nine of the ten classes, showing it had consistently strong performance in identifying most signatures. However, for class 8 (Peak), its accuracy dropped to 0.76 — a rather large drop relative to the other classes. Class 0 (None) was the prediction that was made for the majority of the incorrect predictions for class 8, i.e. the model failed to identify the peak and did not find any other class signatures, so predicted the None class. We used the interpretability facilitated by `MILLET` to investigate what was happening with these incorrect predictions. We found that, in some cases, the model was able to identify the correct region (positive predictions for class 8 at the location of the peak) despite its final prediction being incorrect. Examples are shown in Fig. A.4 — observe how the middle example shows support for class 8 (Peak) in the correct region despite the model getting the overall prediction incorrect. This identification of incorrect predictions and ability to analyse the model's decision-making in more detail further highlights how `MILLET` can be useful in production.

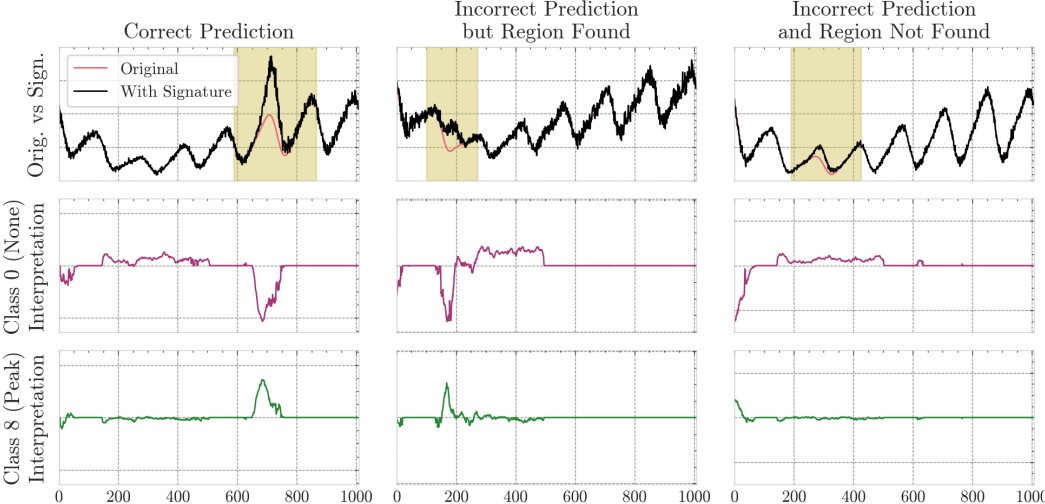

Figure A.4: False negative investigation for `Conjunctive InceptionTime` on the *WebTraffic* dataset. For the final example, the deviation from the original time series is very small and centred near an existing peak, which makes correct identification more difficult. **Top:** The signature time series for an example from class 8 (Peak), where the original time series and signature region are shown. **Middle:** The model's interpretation for class 0 (None). **Bottom:** The model's interpretations for class 8 (Peak).

### E.2 PERFORMANCE BY DATASET PROPERTIES

As discussed in Sec. 5.1, `Conjunctive InceptionTime` had the best performance on balanced accuracy when evaluated over 85 UCR datasets (outperforming SOTA methods such as `HC2` and `Hydra-MR`). To investigate whether this improvement is due to better performance on imbalanced datasets, we assessed balanced accuracy with respect to test dataset imbalance. To measure the imbalance of a dataset, we use normalised Shannon entropy:

$$\text{Dataset Balance} = -\frac{1}{\log c} \sum_{i=1}^{c} \frac{c_i}{n} \log\left(\frac{c_i}{n}\right), \tag{A.11}$$

where $c$ is the number of classes, $c_i$ is the number of time series for class $i$, and $n$ is the total number of time series in the dataset $\left(\sum_{i=1}^{c} c_i = n\right)$. This gives a score of dataset balance between

0 and 1, where 1 is perfectly balanced (equal number of time series per class) and values close to 0 indicate high levels of imbalance. We used a threshold of 0.9 to identify imbalanced datasets, and provide comparisons of `Conjunctive InceptionTime` with GAP `InceptionTime`, `HC2`, and `Hydra-MR` on these datasets in Fig. A.5.

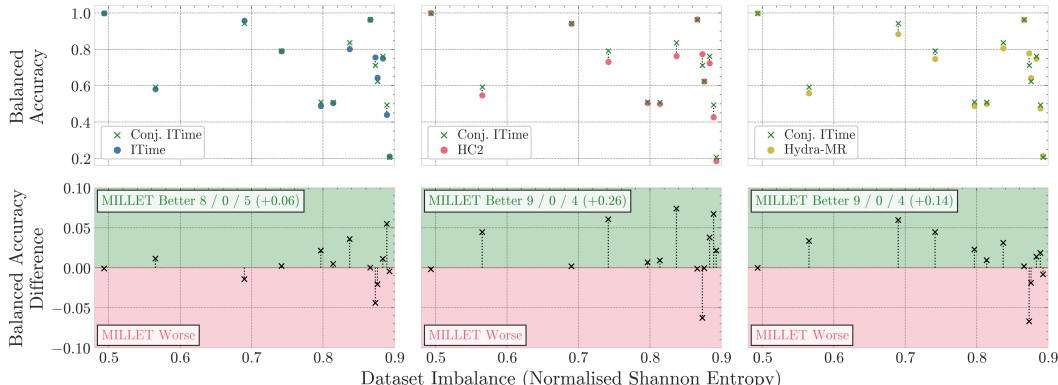

Figure A.5: The effect of dataset imbalance on `MILLET` (`Conjunctive InceptionTime`) compared with GAP `InceptionTime` (left), `HC2` (middle), and `Hydra-MR` (right). **Top:** Balanced accuracy on the imbalanced datasets for `MILLET` and each comparative method. **Bottom:** The gain in balanced accuracy on the imbalanced datasets when using `MILLET`. Numbers indicate `MILLET`'s win/draw/loss and total improvement in balanced accuracy on these 13 datasets.

From these results, we identify that `MILLET` has better performance than the SOTA methods when there is high (test) dataset imbalance. It wins on 8/13, 9/13, and 9/13 datasets against GAP `InceptionTime`, `HC2`, and `Hydra-MR` respectively, and improves balanced accuracy by up to 7.4%. This demonstrates `MILLET` is more robust to dataset imbalance than the other methods, potentially due to the fact that is has to make timestep predictions. Note this is without any specific focus on optimising for class imbalance, e.g. weighted cross entropy loss could be used during training to further improve performance on imbalanced datasets.

Using the UCR results, we now compare performance across time series length and the number of training time series. Figure Fig. A.6 shows the average balanced accuracy rank on different partitions of the UCR datasets for `HC2`, `Hydra-MR`, `InceptionTime`, and `Conjunctive InceptionTime`. The results are relatively consistent across different time series lengths. However, for the number of training time series, we see that `Conjunctive InceptionTime` is worse than GAP `InceptionTime` for smaller datasets, but excels on the larger datasets.

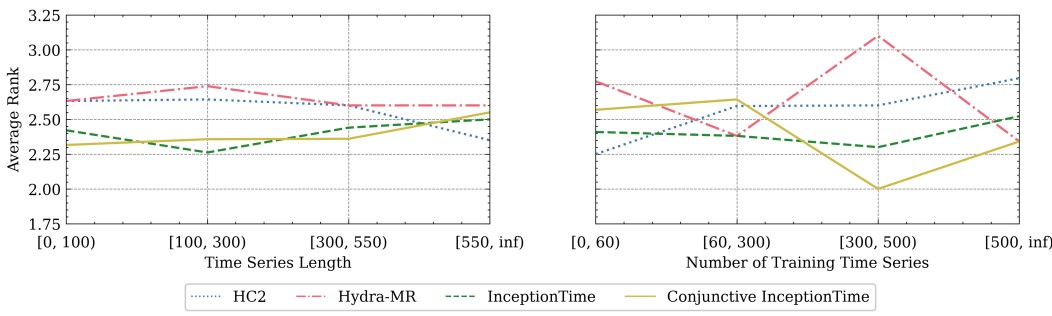

Figure A.6: Comparison of `Conjunctive InceptionTime` against GAP, `HC2`, and `Hydra-MR` for two dataset properties. Bin ranges are chosen to give approx. equal bin sizes.

### E.3 MODEL VARIANCE STUDY

As noted by Middlehurst et al. (2023), while `InceptionTime` performs well overall, it often performs terribly on certain datasets, i.e. it has high variance in its predictive performance. In Fig. A.7, we show that `MILLET` does aid in reducing variance while improving overall performance. Notably, `Conjunctive InceptionTime` has lower variance than `HC2` and `Hydra-MR`.

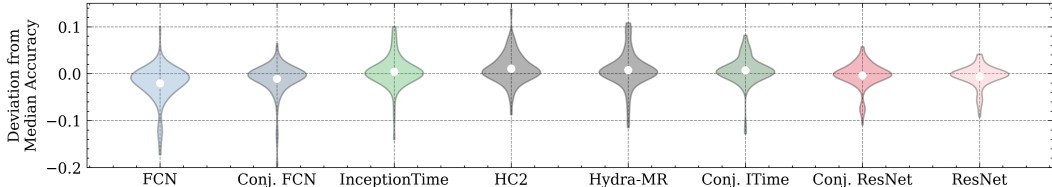

Figure A.7: Evaluation of model variance with respect to median accuracy. Models are ordered from left to right by total variance.

### E.4 RUN TIME ANALYSIS

Below we analyse how `MILLET` increases the complexity of the original backbone models. We first compare the number of model parameters (App. E.4.1) for different backbone and pooling combinations applied to the UCR `Fish` dataset, which is chosen as it is relatively central in the distribution of dataset statistics (175 training time series, 463 time points per time series, and 7 classes). We then compare the training and inference times on this same dataset (App. E.4.2). Finally, we calculate the time complexity of the differnet pooling approaches and assess how they scale with respect to the number of timesteps and the number of classes (App. E.4.3).

#### E.4.1 NUMBER OF PARAMETERS

In Table A.13, we detail the number of model parameters for the different backbones and aggregation approaches on `Fish`. `Instance` has the same number of parameters as GAP as the only change in the aggregation process is to swap the order in which pooling and classification are applied. Similarly, `Attention`, `Additive`, and `Conjunctive` all have the same number of parameters as each other, as they all include the same attention head (just applied in different ways). Including this attention head only leads to an increase of approximately 0.4% in the number of parameters. Note these exact values will change for datasets with different numbers of classes, but the number of additional parameters for the attention head will remain the same.

Table A.13: Number of model parameters for `Fish`. Percentages indicate the relative increase in parameters from the GAP model.

| Pooling | FCN | ResNet | InceptionTime |
|---|---|---|---|
| GAP | 265.6K | 504.9K | 422.3K |
| Attention | 266.6K (+0.4%) | 505.9K (+0.2%) | 423.3K (+0.2%) |
| Instance | 265.6K (+0.0%) | 504.9K (+0.0%) | 422.3K (+0.0%) |
| Additive | 266.6K (+0.4%) | 505.9K (+0.2%) | 423.3K (+0.2%) |
| Conjunctive | 266.6K (+0.4%) | 505.9K (+0.2%) | 423.3K (+0.2%) |

#### E.4.2 TRAINING/INFERENCE TIME

In Table A.14, we compare model training and inference times for `Fish`, and also include how long SHAP takes to run for these models. Due to the additional complexity of these methods (e.g. making time point predictions, applying attention, and using positional embeddings), the training times increase by up to 6%. Similarly, inference time increases by up to 7.5%. For SHAP, generating a single explanation takes 6+ seconds compared to the `MILLET` explanations which are generated as part of the inference step. Using `Conjunctive` as an example, SHAP is over 800 times slower than `MILLET`.

Table A.14: Run time (wall clock) analysis results using `InceptionTime` on UCR `Fish`. Percentages following the `MILLET` methods give the increase in time relative to the backbone GAP model. Note inference time is given in milliseconds but SHAP is given in seconds.

| Model | Train (**seconds**) | Inference (**milliseconds**) | SHAP (**seconds**) |
|---|---|---|---|
| GAP | $565 \pm 2$ | $7.50 \pm 0.02$ | $6.21 \pm 0.03$ |
| Attention | $597 \pm 1$ (+5.7%) | $8.02 \pm 0.02$ (+6.9%) | $6.53 \pm 0.01$ (+5.2%) |
| Instance | $582 \pm 1$ (+3.0%) | $7.75 \pm 0.01$ (+3.3%) | $6.38 \pm 0.02$ (+2.6%) |
| Additive | $599 \pm 1$ (+6.0%) | $8.06 \pm 0.01$ (+7.5%) | $6.51 \pm 0.02$ (+4.8%) |
| Conjunctive | $599 \pm 1$ (+6.0%) | $8.05 \pm 0.01$ (+7.3%) | $6.51 \pm 0.02$ (+4.8%) |

### E.4.3 TIME COMPLEXITY ANALYSIS

To provide more thorough theoretical analysis, we give results for the number of real multiplications in the pooling methods (Freire et al., 2022). We focus solely on the pooling methods, omitting the computation of the backbone (as it is independent of the choice of pooling method). We also omit the computation of activation functions and other non-linear operations. Results are given for single inputs (no batching) and ignore possible underlying optimisation/parallelisation of functions. We first define the number of real multiplications for common functions of the pooling methods:

- Feed-forward layer: $t * d * o$, where $t$ is the number of timesteps, $d$ is the input size (128 in this work), and $o$ is the output size (based on Shah & Bhavsar, 2022).
- Attention head consisting of two layers: $t * d * a + t * a$, where $a$ is the size of the hidden layer in the attention head (8 in this work). This is consistent for the architectures that use attention (`Attention`, `Additive`, and `Conjunctive`); see Tables A.2, A.4 and A.5.
- Weight or average a list of tensors: $n * l$, where $n$ is the number of tensors and $l$ is the length of each tensor.

Given these definitions, we calculate the overall number of multiplications for each pooling method. We provide results in Table A.15 and visualisation in Fig. A.8. We make several observations:

- `Attention` has the best scaling with respect to $t$ and $c$. However, it remains a poor choice overall due to its poor interpretability performance compared to other methods.
- GAP + CAM and `Instance` are the next fastest pooling methods. `Instance` is more efficient than GAP + CAM when $t * c < d * (t + c)$. In this work, as $d = 128$, `Instance` is marginally quicker than GAP + CAM.
- `Additive` and `Conjunctive` are the two slowest methods due to the additional overhead of applying attention and also making instance predictions, but the margin between them and GAP + CAM / `Instance` is not as large as one might expect. This is due to the small hidden layer size in the attention head ($a = 8$). `Conjunctive` is more efficient than `Additive` when $c < d$, which is true in all cases in this work as $d = 128$.
- In general, `Conjunctive` is the best choice when computational cost is less important than predictive performance. If faster computation is required, or if the number of timesteps or number of classes is very large, `Instance` becomes a better choice.

Table A.15: Time complexity analysis (number of real multiplications) of different MIL pooling approaches. For GAP, we include the calculation of CAM as this is required to produce interpretations (the other pooling methods produce interpretations inherently). We also state how the number of real multiplications scale with respect to the number of timesteps $t$ and the number of classes $c$.

| Pooling Method | Number of Real Multiplications | Scale w.r.t $t$ | Scale w.r.t $c$ |
|---|---|---|---|
| GAP + CAM | $d * t * c + d * t + d * c$ | $d * c + d$ | $d * t + d$ |
| Attention | $d * t * a + (2d + a) * t + d * c$ | $d * a + 2d + a$ | $d$ |
| Instance | $d * t * c + t * c$ | $d * c + c$ | $d * t + t$ |
| Additive | $d * t * c + d * t * a + (a + d + c) * t$ | $d * c + d * a + a + d + c$ | $d * t + t$ |
| Conjunctive | $d * t * c + d * t * a + (a + 2c) * t$ | $d * c + d * a + a + 2c$ | $d * t + 2 * t$ |

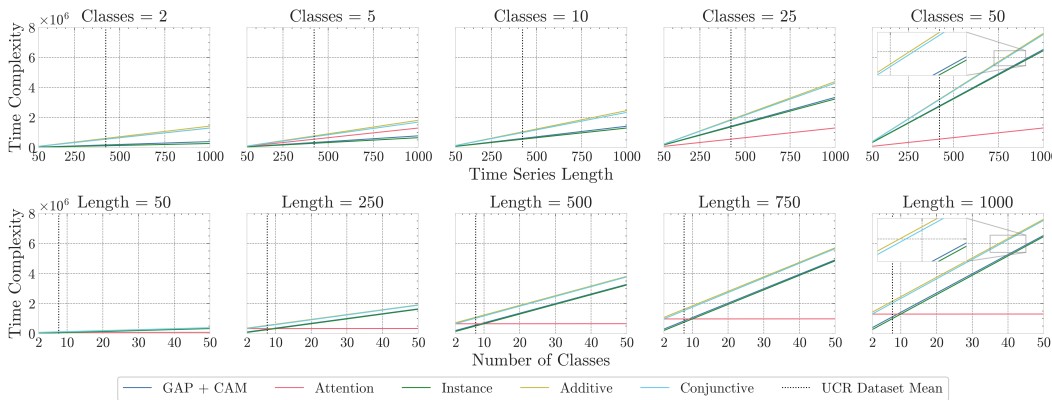

Figure A.8: Visualisation of time complexity (number of real multiplications) for the different pooling methods used in this work. The mean time series length / number of classes across the 85 UCR datasets used in this work is shown (vertical dashed lines). **Top:** Time complexity as time series length increases for varying number of classes. **Bottom:** Time complexity as the number of classes increases for various time series lengths.

### E.5  ABLATION STUDY

Our `MILLET` models make several improvements over the backbones, adding MIL pooling, positional encodings, replicate padding, and dropout (Sec. 3.4). To understand where the gains in performance over the backbone models come from, we conduct an ablation study. To do so, we run additional model training runs, starting with the original backbone models and incrementally adding `MILLET` components in the order: MIL pooling, positional encoding, replicate padding, dropout, and ensembling. The final stage represents the full `MILLET` implementation. To reduce overheads in model training time and compute resources, we focus on `Conjunctive InceptionTime` and conduct the study on the three UCR datasets where `MILLET` shows the biggest increase in balanced accuracy over the backbone: `BeetleFly`, `Lightning7`, and `FaceAll`.

In Table A.16 we provide results of the ablation study for balanced accuracy (predictive performance). We note that, on average, each component of `MILLET` improves performance, and the complete implementation (Step 6) has the best performance. For these datasets, the use of MIL pooling always improves performance over GAP. Additional components then change the performance differently for the different datasets. For example, positional encoding is very important for `BeetleFly`, but replicate padding is most important for `FaceAll`. Interestingly, replicate padding gives the biggest average performance increase across these datasets. However, these results are confounded by the order of implementation, i.e. replicate padding is only applied after MIL pooling and positional encoding have been applied. As such, further studies are required to untangle the contribution of each component, but that is beyond the scope of this work.

Table A.16: Ablation study on balanced accuracy. We begin with the original backbone model (without ensembling). We then incrementally add `MILLET` components until we reach the complete `MILLET` model. Results are given as balanced accuracy / improvement, where improvement is the difference in balanced accuracy relative to the prior row.

| Model Config | BeetleFly | Lightning7 | FaceAll | Mean |
|---|---|---|---|---|
| 1. GAP (Single) | 0.870 | 0.819 | 0.910 | 0.867 |
| 2. + MIL | 0.870 / +0.000 | 0.844 / **+0.024** | 0.912 / +0.002 | 0.875 / +0.009 |
| 3. + Pos Enc | 0.900 / **+0.030** | 0.832 / -0.012 | 0.911 / -0.001 | 0.881 / +0.006 |
| 4. + Replicate | 0.900 / +0.000 | 0.840 / +0.008 | 0.967 / **+0.057** | 0.903 / **+0.022** |
| 5. + Dropout | 0.920 / +0.020 | 0.848 / +0.008 | 0.970 / +0.003 | 0.913 / +0.010 |
| 6. + Ensemble | **0.950** / **+0.030** | **0.863** / +0.015 | **0.974** / +0.003 | **0.929** / +0.016 |

In Table A.17 we provide results of the ablation study for AOPCR (interpretability performance). Interesting, the addition of MIL pooling and positional encoding is detrimental to interpretability in these examples, despite improving predictive performance. However, interpretability improves once replicate padding and dropout are included. Further work is required to understand if these interpretability increases would also occur if replicate padding and dropout were applied to the GAP backbones, or if they improve performance when applied in conjunction with MIL pooling. Finally, we observe that interpretability decreases when ensembling the models. This is somewhat intuitive, as the interpretations are now explaining the decision-making of five models working in conjunction – it would be interesting to explore the difference in interpretations when analysing each model in the ensemble separately rather than together.

Table A.17: Ablation study on AOPCR. Results are given as AOPCR / improvement.

| Model Config | BeetleFly | Lightning7 | FaceAll | Mean |
|---|---|---|---|---|
| 1. GAP (Single) | 0.122 | 4.085 | 0.552 | 1.586 |
| 2. + MIL | -0.736 / -0.858 | 3.985 / -0.100 | 0.859 / +0.306 | 1.369 / -0.217 |
| 3. + Pos Enc | -1.978 / -1.241 | 3.863 / -0.121 | 0.633 / -0.225 | 0.840 / -0.529 |
| 4. + Replicate | -1.926 / +0.052 | 4.093 / **+0.230** | **4.023** / **+3.390** | 2.064 / **+1.224** |
| 5. + Dropout | **1.242** / **+3.168** | **4.253** / +0.159 | 3.997 / -0.026 | **3.164** / +1.101 |
| 6. + Ensemble | 0.787 / -0.456 | 4.190 / -0.063 | 3.585 / -0.412 | 2.854 / -0.310 |

