# OpenReview forum: "Inherently Interpretable Time Series Classification via Multiple Instance Learning"
_ICLR.cc/2024/Conference — ICLR 2024 spotlight_

### Official Review · Reviewer_Yc2u · 2023-10-24

**Soundness:** 3 good
**Presentation:** 3 good
**Contribution:** 3 good
**Rating:** 8
**Confidence:** 4

**Summary:**

This paper introduces a new framework that leverages Multiple Instance Learning to make deep learning TSC models inherently interpretable without compromising predictive performance. The authors evaluate MILLET on 85 UCR TSC datasets and show that it produces sparse explanations quickly and of higher quality than other interpretability methods.

**Strengths:**

The strengths of this paper include:

1.	Introducing a new MILLET framework that makes deep learning TSC models inherently interpretable by leveraging the MIL approach without compromising predictive performance. In particular, the authors proposed exploring four MIL (attention, instance, additive and conjunctive) pooling methods to increase interpretability while replacing GAP. Moreover, including positional encoding ensures the modelling of time series constraints.
2.	Proposing a new synthetic dataset called WebTraffic to explore the MILLET concept and evaluate the inherent interpretability of their models. The authors compared the four proposed MIL pooling approaches for MILLET with GAP on their WebTraffic dataset. Each pooling method is applied to the FCN, ResNet, and InceptionTime backbones.
3.	Evaluating MILLET on 85 UCR datasets and showing that it produces sparse explanations quickly and of higher quality than other interpretability methods. The authors found that
while Conjunctive InceptionTime is the best approach for balanced accuracy (outperforming the HC2 and Hydra-MR SOTA methods), it is not quite as strong on the other metrics. However, it remains competitive, and for each backbone, using MILLET improves performance across all metrics. Moreover, the authors found that the Conjunctive has the best interpretability performance.

**Weaknesses:**

The major weaknesses are summarized below:

1.	The paper does not provide a detailed comparison of MILLET with other state-of-the-art TSC models. Although the authors claim that “We design three MILLET DL models by adapting existing backbone models that use GAP: FCN, ResNet, and InceptionTime. While extensions of these methods and other DL approaches exist (see Foumani et al., 2023), we do not explore these as none have been shown to outperform InceptionTime (Middlehurst et al., 2023).” the application of MILLET to other models and further comparisons with other state of the art TSC is missing and is relevant better to measure the effectiveness and generalizability of the proposed approach.

2.	The paper does not provide a detailed comparison with other TSC interpretability methods. (e.g.LIME)

3.	The paper does not provide a detailed algorithm complexity analysis. While the authors provide information on the run time of MILLET (see E.3), a more detailed analysis of the algorithm complexity would help establish its scalability and feasibility in large-scale applications.

**Questions:**

1.	How does MILLET perform with other TSC models?
2.	How does MILLET compare with other TSC interpretability methods?
3.	Can you provide a more detailed analysis of the algorithm complexity?
4.	Have you considered the potential impact of the choice of hyperparameters on the performance of MILLET?
5.	Have you considered the potential impact of class imbalance on the performance of MILLET?

After reading the author's rebuttal and discussions I am more incline to accept the paper.

---

> ### Author Response · Authors · 2023-11-17
> **Response to Reviewer Yc2u**
>
> Thank you for your time in reviewing our work. Please see our response to your review below in addition to the general rebuttal given above.
>
> 1. *Application of MILLET to other TSC methods (Weakness 1 / Question 1)*
> In the work we focused on applying MILLET to the deep learning family of TSC methods. As stated in the paper, “we choose to focus on DL approaches due to their popularity, strong performance, and scope for improvement” (Section 2). We agree that extending MILLET to other families of TSC methods is a worthwhile area of further study, but believe that our analysis against the current versions of SOTA methods is sufficiently comprehensive to demonstrate the effectiveness of MILLET. While MILLET should still be applicable to other TSC methods, it is not as trivial as applying it to other deep learning methods and warrants standalone future work. We feel that our time during the rebuttal is best served delivering more insights into MILLET for deep learning methods, as this was the main focus of the work.
> 2. *Other interpretability methods (Weakness 2 / Question 2)*
> Thank you for the suggestion of comparing with LIME. With LIME, we expect the analysis to uncover a similar issue as to what we discovered with SHAP: LIME will be very expensive to run as it makes many repeated forwarded passes of the model. We will include a discussion of the alternative interpretability methods in a new section in the Appendix.
> 3. *Time complexity analysis (Weakness 3 / Question 3)*
> We have begun conducting time complexity analysis of the pooling methods, and will expand Appendix E.3 to include these results. This will go beyond run time and number of forward passes, rather looking at actual time complexity and how the pooling methods scale with the number of classes and time series length. Please see the general rebuttal for initial results.
> 4. *Impact of hyperparameter choice (Question 4)*
> As discussed in Appendix B.2, we chose fixed hyperparameter values based on those used to train the original backbone models (i.e. no hyperparameter tuning was used). This was done to ensure a fair comparison and provide a set of default hyperparameter values for use in derivative works. We will happily expand Appendix B.2 to mention the impact of hyperparameter choice. For example, MILLET had varying convergence rates for different datasets: sometimes it converged very quickly (so a lower LR might be better), but sometimes it took a long time to converge, so more epochs may be required. Scheduled/adaptive LR might be appropriate in some cases.
> 5. *Impact of class imbalance (Question 5)*
> As mentioned in your review, MILLET was the best approach in our evaluation of balanced accuracy performance. This suggests it is more robust to class imbalance than other methods. This was also without any specific focus on optimising for class imbalance, e.g. weighted cross entropy loss could be used during training to further improve performance on imbalanced datasets. However, further analysis of the effect of class imbalance would indeed be insightful. We will expand Appendix E.1 to include this. Our initial results have shown that performance improvements are demonstrated for MILLET on datasets with a high level of imbalance compared to InceptionTime, HC2, and Hydra-MR.

---

> > ### Comment · Reviewer_Yc2u · 2023-11-18
> > **Response to authors**
> >
> > I am satisfied with the authors' response. The focus on hyperparameters, time complexity and discussing the impact of class imbalance should have priority over the application of MILLET to other TSC ML approaches that may require future work. However, the hyperparameter tuning may influence the performance of the proposed approach. The authors should provide evidence of that.
> > Based on the time complexity analysis, I expect to find a general discussion summarising the best choice (on average) as a trade-off between computation effort and predictive performance.

---

> > > ### Author Response · Authors · 2023-11-21
> > >
> > > Thank you for your response to our rebuttal. We have now uploaded a new version and welcome your thoughts on the new experiments/discussion of hyperparameters, time complexity, and dataset imbalance.

---

> > > > ### Comment · Reviewer_Yc2u · 2023-11-22
> > > > **Re-response**
> > > >
> > > > Although the authors did not provide a sensitive analysis of hyperparameters, I am still satisfied with the author's response.

---

### Official Review · Reviewer_uQF7 · 2023-10-25

**Soundness:** 3 good
**Presentation:** 3 good
**Contribution:** 3 good
**Rating:** 8
**Confidence:** 4

**Summary:**

The paper presents MILLET a model for Multiple Instance Learning for Locally Explainable Time series classification.

**Strengths:**

+ The paper is well written and all the choices are justified and carefully explained
+ The experimentation is deep
+ The proposal is novel (to the best of my knowledge)
+ The references are updated

**Weaknesses:**

- I would have appreciated a comparison with LIME or with LIMESegments
- I would have appreciated a comparison against ROCKET or MiniROCKET at least as competitor for the TSC task. Further usage of MILLETS also for ROCKET will completely fulfill the purpose of proposing this approach as a model-agnostic one.
- To fully understand the paper the reader is constrained to refer to the Supplementary Material. A suggestion is to save some space and anticipate in the main paper some of the details of the Supplementary Material.
- Experiments with the synthetic dataset should have been performed by varying the number of records, time stamps, classes.

**Questions:**

Questions can be derived from the weaknesses part.

---

> ### Author Response · Authors · 2023-11-17
> **Response to Reviewer uQF7**
>
> Thank you for your time in reviewing our work. Please see our response to your review below in addition to the general rebuttal given above.
>
> 1. *Other interpretability methods (Weakness 1)*
> Thank you for the suggestion of comparing with LIME and LIMESegments. For LIME, we expect the analysis to uncover a similar issue as to what we discovered with SHAP: LIME will be very expensive to run as it makes many repeated forward passes of the model. We will include a discussion of the alternative interpretability methods in a new section in the Appendix.
> 2. *Application of MILLET to ROCKET (Weakness 2)*
> In the work we focused on applying MILLET to the deep learning family of TSC methods. As stated in the paper, “we choose to focus on DL approaches due to their popularity, strong performance, and scope for improvement” (Section 2). We agree that extending MILLET to other families of TSC methods is a worthwhile area of further study, but believe that our analysis against the current versions of SOTA methods is sufficiently comprehensive to demonstrate the effectiveness of MILLET. Convolutional methods such as ROCKET often use a different form of pooling to deep learning methods: while deep learning employs GAP, convolutional models use alternative approaches such as max pooling and proportion of positive values (PPV) pooling. While MILLET should still be applicable to these methods, it is not as trivial as applying it to other deep learning methods and warrants standalone future work. We feel that our time during the rebuttal is best served delivering more insights into MILLET for deep learning methods, as this was the main focus of the work.
> 3. *Understanding requires supplementary material (Weakness 3)*
> We appreciate there are a lot of references to the appendix throughout the work. We attempted to signpost the relevant appendices as much as possible to aid clarity and indicate where readers should look for further detail. We are happy to improve the clarity of the main body in our revised submission.
> 4. *Synthetic dataset should have varied the number of records, etc. (Weakness 4)*
> One of the strengths of our synthetic dataset (WebTraffic) is indeed that it facilitates precise control of dataset properties (dataset size, time series length, number of classes, etc.). However, in this work we relied on the 85 UCR datasets to cover a broad range of TSC domains and properties, but WebTraffic could certainly be used for this purpose. We hope that it may be used in derivative works to further investigate interpretable TSC.

---

> > ### Comment · Reviewer_uQF7 · 2023-11-22
> >
> > Thank you for your reply.

---

### Official Review · Reviewer_pSU6 · 2023-10-30

**Soundness:** 4 excellent
**Presentation:** 4 excellent
**Contribution:** 4 excellent
**Rating:** 8
**Confidence:** 4

**Summary:**

The paper introduces a novel intrinsically interpretable deep learning model. The authors framed time series classification as a Multiple Instance Learning (MIL) which can highlight the most influential time points in the outcome of the model. This method employs a various techniques such as attention, instance pooling, additive pooling, and conjunctive pooling across an ensemble of deep methods where each method offer different interpretability.

**Strengths:**

Nicely written. Well evaluated. Novelty.

**Weaknesses:**

I was not able to identify any weakness.

**Questions:**

Overall, this paper could be a significant algorithmic contribution and I think the authors done amazing job on presenting it. I wonder if the method can be applied to other domains.

---

> ### Author Response · Authors · 2023-11-17
> **Response to Reviewer pSU6**
>
> Thank you for your time in reviewing our work. Please see our response to your review below in addition to the general rebuttal given above.
>
> We believe there could be some scope for applying our work in other machine learning paradigms (we assume this the ‘other domains’ that you are referring to – please correct us if this is not the case!). The existing pooling methods we study (instance, attention, and additive) are applicable in other machine learning paradigms (e.g. MIL for vision), so our novel pooling method (conjunctive) should also be applicable elsewhere. We will mention this in a new appendix section.

---

> > ### Comment · Reviewer_pSU6 · 2023-11-22
> >
> > That was exactly what I meant. I check other reviews and your response. In my humble opinion this is a great paper and deserve to be publish.

---

> > > ### Author Response · Authors · 2023-11-23
> > >
> > > We are glad that our response aligned with your request. Thank you very much for your kind words and feedback on our work.

---

### Official Review · Reviewer_NeUd · 2023-11-01

**Soundness:** 3 good
**Presentation:** 3 good
**Contribution:** 3 good
**Rating:** 8
**Confidence:** 3

**Summary:**

This paper introduces a framework based on Multiple Instance Learning to enhance the interpretability of time series classification models. The framework, MILLET, proposes passing the extracted feature embeddings from any backbone (e.g. FCN, ResNet) through a positional encoding, dropout, and a final pooling layer. Depending on the pooling structure, the method can make the underlying model more interpretable in some settings while also improving performance in others. The authors propose a new pooling method, conjunctive pooling, specifically for time series. MILLET is evaluated through the construction of 12 new models on 85 UCR datasets and a newly introduced synthetic dataset for interpretability evaluation. This approach represents a novel application of MIL to TSC and offers improved interpretability in various domains.

**Strengths:**

Thanks to the authors for their submission: it contains useful research that shows good research practices while explaining an interesting and novel idea within multivariate time series classification and interpretability. The results of this work will be informative to other researchers and are significant in improving our understanding of applying deep learning methods with time series. Some specific strengths of this research:

- The Multiple Instance Learning presented in this work has more general applications within time series classification than previous work and provides a more robust evaluation of the benefits and drawbacks across a range of tasks, both synthetic and real.
- MILLET model design adds very little complexity to existing models while contributing improved interpretability. It is flexible enough to work with any backbone model (FCN, ResNet, InceptionTime, and more) while maintaining performance.
- The proposed synthetic dataset, WebTraffic, provides a helpful contribution to the task of benchmarking time series interpretability. With the ability to scale up to large sizes and the replication of a common time-series use-case it could be a helpful foundation to build-on in the future.
- Performance of Conjunctive Pooling shows improvements across the class of neural network models for time series classification.

**Weaknesses:**

While the results and paper are generally strong, there are a few areas for improvement particularly as regards to interpretability claims:

- Lack of comparison against previous benchmarks for saliency maps for feature attribution in time series classification from TSR [1], DynaMask [2], WinIT [3], and TimeX [4]. All of these works provide additional synthetic datasets for evaluating interpretability methods and show performance against more general methods like Feature Occlusion and Integrated Gradients. While MILLET seems like to improve on such methods due to the better computational efficiency, it is not thoroughly evaluated in the paper, except in counting the number of forward passes in the difference between SHAP, CAM, and MILLET.

- Interpretability evaluation metrics. It is not clear that AOPCR and NDCG@n can be strictly ported over to the time-series setting. For example, as pointed out, with NDCG@n the time points in the middle of a region of missing data may be considered important by the ground truth, but may not be highlighted by the interpretability method, instead the beginning or end may be highlighted. These scores may be weighted differently for similar outcomes. More discussion around the impact of this is relevant to researchers.

[1] https://arxiv.org/pdf/2010.13924.pdf
[2] https://arxiv.org/pdf/2106.05303.pdf
[3] https://arxiv.org/pdf/2107.14317.pdf
[4] https://arxiv.org/pdf/2306.02109.pdf

**Questions:**

These questions will help clarify my understanding of the paper. Some of these could benefit from additional analysis in the paper itself:

1/ What is the author’s intuition for the added performance of conjunctive pooling over other pooling methods?
2/ In Figure 6, are the x’s referring to different pooling methods for MILLET or multiple runs of the Conjunctive pooling model?
3/ One of the most interesting things about the WebTraffic dataset is the different signatures grounded in real-world patterns. The authors note that the Conjunctive InceptionTime model identifies regions around Spikes and only the start and end of Cutoffs. Does classifier or pooling selection change how the interpretability functions or performs across these various class types? Can this tell us anything more about how the Conjunctive Pooling functions or why it performs better?

---

> ### Author Response · Authors · 2023-11-17
> **Response to Reviewer NeUd**
>
> Thank you for your time in reviewing our work. Please see our response to your review below in addition to the general rebuttal given above.
>
> 1. *Other interpretability methods (Weakness 1)*
> Thank you for the references to additional interpretability metrics for TSC. We will include a discussion in a new appendix section contrasting our approach with these methods.
> 2. *Time complexity analysis (Weakness 1 cont.)*
> We have begun conducting time complexity analysis of the pooling methods, and will expand Appendix E.3 to include these results. This will go beyond run time and number of forward passes, rather looking at actual time complexity and how the pooling methods scale with the number of classes and time series length. Please see the general rebuttal for initial results.
> 3. *Interpretability evaluation metrics (Weakness 2)*
> We will include a discussion of additional evaluation metrics in a new appendix section and contrast them with AOPCR and NDCG@n.
> 4. *Intuition as to why conjunctive pooling is more effective (Question 1)*
> In Section 3.2, when we define conjunctive pooling, we state “… [conjunctive pooling] is expected to benefit performance as the attention and classifier heads are trained in parallel rather than sequentially, i.e. the classifier cannot rely on the attention head to alter the time point embeddings prior to classification, making it more robust.” Through our new time complexity analysis, we find that conjunctive pooling is also more efficient than additive pooling (the most similar pooling method) when the number of classes is less than the size of the extracted features. This is always true in this work as the feature embeddings are of length 128.
> 5. *Figure 6 (Question 2)*
> Each x in Figure 6 refers to a different pooling method (Instance, Additive, or Conjunctive). Attention pooling is omitted from this diagram as it performs poorly. We will happily modify the caption to clarify this.
> 6. *Performance across classes in WebTraffic (Q3)*
> Through analysis of the confusion matrix for Conjunctive InceptionTime on the WebTraffic dataset, we found that the model has relatively consistent performance across all classes, i.e. there isn’t one particular class where it fails drastically compared to the other classes. The class where it does the worst is the peak class; occasionally missing peaks (identifying them as class 0 – None). Further investigation found the interpretations are sometimes still able to identify the correct region, despite the prediction being wrong. This suggests conjunctive pooling is useful for investigating incorrect predictions. We will include an additional appendix section that provides this analysis.

---

> > ### Comment · Reviewer_NeUd · 2023-11-22
> >
> > Thanks to the authors for their response to our reviews. The responses helped answer the three questions I highlighted.
> >
> > In response to Q2, I appreciate the clarification. It is not clear to me why Attention pooling is omitted simply because of poor performance? Perhaps this result should be included as it contrasts with the similar (even best accuracy) performance to other pooling methods on the WebTraffic dataset as shown in the appendix.
> >
> > The response to Q3, and the updated appendix section on conjunctive pooling's performance with the WebTraffic dataset was useful in bringing to light some of the nuances of interpreting the results of these interpretability methods. The separation of examples of finding correct regions with incorrect predictions and the completely incorrect predictions was interesting, and as mentioned, suggests the method could be useful for conducting such investigations and there may be other evaluation metrics to capture this dynamic.
> >
> > I appreciate the authors time in conducting the time complexity analysis and adding it to appendix E.3, as well as the discussion around evaluation.

---

> > > ### Author Response · Authors · 2023-11-22
> > >
> > > Thank you for the ongoing discussion. We are glad that our response has helped answer your questions.
> > >
> > > For Figure 6, Attention is omitted purely to aid visualisation. This plot is a representation of the data presented in tables A.10 and A.11 (Appendix D.3). As the interpretability performance of attention is significantly worse than the other approaches, including it in the plot makes visualisation more difficult given the limited space. The aim of this plot is to convey the trade-off between interpretability and predictive performance, which we feel is achieved without including the results for attention pooling.

---

### Author Response · Authors · 2023-11-17
**General Rebuttal**

We would like to firstly thank all the reviewers for their time spent reviewing and providing feedback on our work. The comments are very thorough and provide insight into how we can improve the paper. Reviewers highlighted the specific contributions of our MILLET approach: it improves interpretability without sacrificing predictive performance and comes with little additional computational overhead. The usefulness of our novel synthetic dataset (WebTraffic) was also acknowledged. Reviewers agreed that the work was novel, well-written, and rigorously evaluated.

Below we outline points for discussion that were raised by multiple reviewers. For reviewer-specific comments, please see our responses to each individual review. We then propose a set of planned changes to improve the manuscript based on reviewer feedback. We hope these comments help facilitate an ongoing discussion with all reviewers.

**Reponses to Common Reviewer Feedback:**
1. *Lack of comparison against other interpretability methods, and motivation for choice of interpretability metrics.*
Several reviewers stated it would be beneficial to compare MILLET against additional TSC interpretability methods. We will provide a discussion   in the Appendix contrasting these methods with our approach, and also our choice of AOPCR and NDCG@n with alternative evaluation metrics. Our findings thus far indicate that MILLET and our chosen evaluation method overcomes several problems with existing work, most notably with regards to computational complexity (MILLET is much faster and is able to scale to long time series).
2. *Extension of MILLET to additional TSC methods.*
In the work we focused on applying MILLET to the deep learning family of TSC methods. As stated in the paper, “we choose to focus on DL approaches due to their popularity, strong performance, and scope for improvement” (Section 2). Furthermore, we mentioned that future work could focus on extending MILLET to other families, for example convolutional (e.g. Rocket)   methods. However, these methods often use a different form of pooling to deep learning methods: while deep learning employs GAP, convolutional models use alternative approaches such as max pooling and proportion of positive values (PPV) pooling. While MILLET should still be applicable to these methods, it is not as trivial as applying it to other deep learning methods and warrants standalone future work. We agree that this is a worthwhile area of further study, but believe that our analysis against the current versions of SOTA methods is sufficiently comprehensive to demonstrate the effectiveness of MILLET.
3. *More thorough evaluation of computational complexity required.*
While we analysed the model size and run time of MILLET in Appendix E.3, reviewers suggested further analysis of the computational complexity of the method would be beneficial and could be used to determine the feasibility of MILLET for large scale applications. We will update the appendix with a more in-depth discussion of the time complexity of the different pooling approaches. Our initial findings are included as a comment to this response (due to OpenReview character limits).

We feel that our time during the rebuttal is best served in delivering more insights into the performance of MILLET for deep learning methods (e.g. analysing the computational complexity, robustness to dataset imbalance, and why conjunctive pooling is effective), rather than expanding the work to include more interpretability baselines and evaluation metrics. Exploring the proposed methods in more depth offers further clarity about why MILLET is beneficial.

**Planned Changes**
1. Further analysis on the time complexity of the different pooling approaches (Appendix E.3).
2. Additional analysis on the effect of dataset imbalance (Appendix E.1).
3. New section in Appendix E on the effectiveness of Conjunctive pooling.
4. New appendix section on further interpretability baselines and evaluation metrics.
5. Inclusion of possible extensions of our work (specifically conjunctive pooling) to other machine learning paradigms (e.g. vision and video).
6. Expand hyperparameter discussion (Appendix B.2) to include the potential impact of changing the hyperparameter values.
7. Modify the caption of Figure 6 for additional clarity around what each x represents.

Thank you once again for your time spent reviewing our work – we greatly appreciate your comments.

---

> ### Author Response · Authors · 2023-11-17
> **Initial Time Complexity Analysis**
>
> In this analysis, we calculated the number of real multiplications [1] of the five different pooling methods (GAP, Attention, Instance, Additive, and Conjunctive). This omits the time complexity of the backbone, which is equal in all cases. We also only focused on the core operations of the different pooling approaches and omit the computation of activation functions and other non-linear operations. Results are given for single inputs (no batching) and ignore underlying optimisation/parallelisation of functions.
>
> Variables:
> $t$ – Number of timesteps.
> $d$ – Dimensionality of features extracted by the backbone feature extractor. In this work, 128 for all backbones.
> $c$ – Number of classes.
> $a$ – Attention head hidden size. In this work, 8 for all pooling methods that use attention (Attention, Additive, and Conjunctive).
>
> Results:
> | Pooling Method | Time Complexity (Number of Real Multiplications) |
> | --------------------- | ---------------------- |
> | GAP + CAM      | $d * t * c +  d * t + d * c$ |
> | Attention           | $d * t * a + (2d + a) * t + d * c$ |
> | Instance            | $d * t * c + t * c$ |
> | Additive            | $d * t * c + d * t * a + (a + d + c) * t$ |
> | Conjunctive     | $d * t * c + d * t * a + (a + 2c) * t$ |
>
> Further analysis and visualisation of method scalability with respect to time series length $t$ and the number of classes $c$ will be given in the revised version of the manuscript.
>
>
> [1] https://arxiv.org/abs/2206.12191

---

### Author Response · Authors · 2023-11-21
**New version uploaded**

We have now uploaded a new version of paper based on reviewer comments. Changes are highlighted in blue to help make them clear. Thank you once again for your comments and ongoing discussions.

**Changes**

Main body:
1. Updated Figure 6 caption for clarity.
2. Mentioned that conjunctive pooling is applicable to general MIL problems as future work.
3. Added references to new appendix sections.

Appendix:
1. A.1: Related Work (Continued) section discussing alternative TSC interpretability methods and metrics.
2. B.2: Expanded discussion of MILLET hyperparameter selection and its potential impact on performance.
3. E.1: Investigation of false negatives using Conjunctive InceptionTime as an example. Explores the additional benefits of MILLET and provides performance across classes in WebTraffic.
4. E.2: Inclusion of performance w.r.t dataset imbalance, showing MILLET outperforms SOTA on imbalanced datasets.
5. E.4.3: Time complexity analysis. Includes visualisations.

---

### Public Comment · ~Weijia_Zhang2 · 2023-11-21
**Interesting new application of MIL, but there may be a discrepency between attention-based pooling and the basic underlying assumption**

Dear Authors,

I extend my sincere appreciation for your thought-provoking paper on integrating MIL into time-series treatment classification. After reading your paper, I would like to present a couple of observations and inquiries for clarification:

In the "Why MIL" section, it is discussed that "Our underlying assumption is that, for a classifier to predict a certain class for a time series, there must be one or more underlying motifs", and this is indeed equivalent to the essence of the \textbf{standard multi-instance assumption} (a positive bag is positive if and only if it has one or more positive instances). However, have the authors considered that all MIL methods that use attention-based pooling actually \textbf{do not satisfy the standard MIL assumption}, as recently discussed in [1]?

[1] Edward Raff and Jim Holt. Reproducibility in Multiple Instance Learning: A Case For Algorithmic Unit Tests. NeurIPS 2023. https://openreview.net/forum?id=aZ44Na3l9p

---

> ### Author Response · Authors · 2023-11-22
>
> Thank you very much for the comment.
>
> As stated in our Background and Related Work, "...we do not constrain [MILLET] to any specific MIL assumption except that there are temporal relationships...". This was due to the large variety of TSC datasets used in the work, with varying numbers of classes and different types of motifs and interactions between those motifs that define the classes. As such, the standard MIL assumption is not guaranteed to hold for every TSC dataset we used, especially considering many of the datasets are non-binary (more than two classes).
>
> As the work of [1] focuses solely on SMIL problems, which is an assumption we are not using, it is difficult to apply it to our work. We have conducted "in situ" evaluation (using the terminology of [1]) across a broad range of datasets, whereas [1] is only applicable in specific (binary) "in vivo" evaluations. While this might be applicable for specific applications of MILLET (e.g. those adhering to the SMIL assumption), we do not consider it a relevant (or feasible) evaluation for this work.
>
> Rather than use the notion of binary positive instances in the standard MIL assumption, we use the idea of discriminatory and non-discriminatory instances [2,3] and that discriminatory instances (or motifs) can support or refute different classes. This is generally applicable to all TSC problems and not constrained to binary problems (which SMIL and [1] are).
>
> For the use of attention-based pooling, in Q3 at the end of the "Why MIL" section, we state "...attention is far less explicit than time point predictions – there is no guarantee that attention actually reflects the underlying motifs of decision-making...", which echoes the ideas of [1]. Furthermore, we do not advocate for the use of attention on its own. Our experiments showed that attention pooling has poor interpretability performance, and we suggest it should not be used.
>
> For the two other pooling approaches in our work that use attention (Additive and Conjunctive), it is not clear from [1] whether these have the same issues as attention pooling (violating MIL assumptions), as they are effectively a combination of Instance pooling (mi-Net) and attention. Note [1] did not assess additive pooling despite its strong performance.
>
> Of the different pooling approaches used in our work, we advocate for the use of Instance (mi-Net) and Conjunctive pooling (see Appendix E.3.4), which aligns with the suggestion of [1] to use mi-Net. While [1] also supports the use of CasualMIL [4], this approach is only applicable to (binary) SMIL problems. As we are exploring many different types of TSC datasets that have more than two classes, CausalMIL is not readily applicable to our work.
>
> [1] Edward Raff and Jim Holt. Reproducibility in Multiple Instance Learning: A Case For Algorithmic Unit Tests. NeurIPS (2023)
> [2] Amores, Jaume. Multiple instance classification: Review, taxonomy and comparative study. Artificial intelligence (2013)
> [3] Joseph Early, Christine Evers, and Sarvapali Ramchurn. Model agnostic interpretability for multiple instance learning. ICLR (2021)
> [4] Zhang, Weijia, Xuanhui Zhang, and Min-Ling Zhang. Multi-instance causal representation learning for instance label prediction and out-of-distribution generalization. Advances in Neural Information Processing Systems (2022)

---

> > ### Public Comment · ~Weijia_Zhang2 · 2023-11-23
> >
> > Thanks for the discussion!
> >
> > Following up on the Additive and Conjunctive pooling, it is my understanding that the instance pooling component is not the same as mi-Net. In mi-Net, the pooling is straightforwardly a **maximum** operation (https://github.com/yanyongluan/MINNs). However, Eq 3 presents the instance pooling as an **average** operation. Actually, Eq 3 in this manuscript is the average pooling (in Eq 5 of Wang et al. 2018.) It seems unlikely that the Additive and Conjunctive pooling will avoid the same issues as the attention pooling. Also, there appears to be no E.3.4 (at least in the PDF available to the public)?
> >
> > This discussion is solely intended to explore the contents of the work; it's not meant as criticism by any means. I do not presume to evaluate the overall merit of this paper as I was not the reviewer. If I were the reviewer, I would also recommend acceptance based on effectively exploring a novel and important application for MIL.

---

### Meta-Review · Area_Chair_j8QG · 2023-12-05

**Metareview:**

The paper proposes MILLET, a novel framework that leverages Multiple Instance Learning (MIL) to make deep learning models for time series classification (TSC) inherently interpretable. It offers several strengths:

- Novelty: Introduces a novel approach for interpretable TSC, with several important components (attention, instance, additive, conjunctive).
- Improved Interpretability: Provides sparse and high-quality explanations compared to other methods, as demonstrated.
- Performance: Maintains or improves performance compared to strong alternatives, demonstrating its effectiveness.
- Flexibility: Works with various backbone models, making it applicable to existing models.
- Evaluation: Thoroughly evaluated on synthetic and real datasets, demonstrating its generalizability.

Overall, all reviewers are aligned on the acceptance of the paper. Please improve the writing and integrate the proposed changes for the final version.

**Justification For Why Not Higher Score:**

- Limited Comparisons: More comparisons with the state of the art can be added
- Insufficient analysis of algorithm complexity
- Paper writing and figures can be improved

**Justification For Why Not Lower Score:**

- Novelty: Introduces a novel approach for interpretable TSC, with several important components (attention, instance, additive, conjunctive).
- Improved Interpretability: Provides sparse and high-quality explanations compared to other methods, as demonstrated.
- Performance: Maintains or improves performance compared to strong alternatives, demonstrating its effectiveness.
- Flexibility: Works with various backbone models, making it applicable to existing models.
- Evaluation: Thoroughly evaluated on synthetic and real datasets, demonstrating its generalizability.

---

### Decision · Program_Chairs · 2024-01-16

Accept (spotlight)